# Soft matrix promotes immunosuppression in tumor-resident immune cells via COX-FGF2 signaling

Aino Peura [1], Rita Turpin [1,2,13], Ruixian Liu [1,13], Maria Heilala [3], Maria Salmela[4], July Aung [1], Piia Mikkonen[5], Minna Mutka [6], Panu E. Kovanen[6], Laura Niinikoski [7], Tuomo Meretoja[7], Johanna Mattson [8], Päivi Heikkilä[6], Päivi Palanne[9], Tiina Kantanen[6], Mikko Kilpeläinen[9], Outi Ukkonen[9], Maija Hollmén [2], Topi A. Tervonen[1,4], Juha Klefström [1,10,11,12,14] ✉ & Pauliina M. Munne [1,14] ✉

Mechanical forces of the tumor microenvironment change dynamically during key events of tumorigenesis such as invasion and metastasis. These changes in compressive forces often affect the breast cancer cell phenotype. However, it is lesser known how these dynamic mechanical forces in the tumor micro-environment affect the phenotypes of tumor infiltrated leukocytes (TIL) and their subsequent anticancer activities. Here we find, in primary patient-derived explant cultures (PDEC) containing resident TILs, that low compression pro-motes a change in the original identity of breast cancer cells from luminal to a more mesenchymal and undifferentiated state. These altered tumor cells induce an upregulation of immunosuppressive cytokines such as interleukin-10 (IL-10) and Transforming Growth Factor Beta (TGF-β), as well as polarization of macrophages towards pro-tumor M2(Gc)-type and depletion of CD8+ effector memory T-cells. These immunosuppressive events are mediated by tumor cell derived fibroblast growth factor 2 (FGF2) and prostaglandin E2 (PGE2). We also find that FGF2 rich areas in primary tumors show enrichment in M2-like-macrophages and diminished numbers of CD8 + T and B-cells. Our results suggest that low compressive forces in the tumor microenvironment induce local immunosuppression via FGF2 secretion arising from phenotypic plasticity of tumor cells.

The tumor microenvironment (TME) consists of tumor adjacent immune cells, stromal cells, blood vessels, and the extracellular matrix (ECM). Chemical and physical changes within the TME shape tumor growth, progression and invasion[1,2]. While the chemical signaling between tumor and the TME occurs via a variety of growth factors, cytokines, chemokines as well as other chemical ligand-receptor systems, the physical signaling occurs via dynamic changes in the ECM stiffness, liquid pressure, and cell density that are sensed by the tumor cells by a variety of mechanoreceptors and sensors, such as integrins and associated signaling molecules like YAP[2–4].

Many studies have established that tumorigenesis and tumor progression are associated with ECM stiffening through a process in which the tumor and stromal cells deposit, contract, and crosslink the matrix[5]. Besides tumor and stromal cells, emerging evidence shows that the tumor infiltrated leukocytes (TIL) in TME can also sense changes in the mechanical microenvironment[6,7]. Moreover, a stiff

microenvironment has been shown to restrict the access of TILs to tumor tissue and thus possibly promoting resistance to several immunomodulatory therapeutic agents[8]. While tumors undergo global stiffening during the tumor growth and progression, it is important to note that key events of tumorigenesis can cause local softening of the TME. For example, rapidly growing tumors often spontaneously develop hypoxic regions, which can turn into large necrotic areas typically inside of the tumor mass[9]. In addition, cytotoxic chemotherapy can necrotize large parts of the tumor[10]. Tumor cells and tumor resident leukocytes are exposed to sudden or gradual softening of the tumor microenvironment, for example during necrosis, or even different gradients of stiffness over the course of tumor progression. Furthermore, the process of invasion involves the transition of cells from the compact epithelial tumor tissue through the basement membrane to the surrounding stroma. The invasion process involves rapid changes in matrix stiffness that are sensed by invading cells[11]. Thus, local matrix softening coincides with many critical steps of tumorigenesis.

Decreased matrix stiffness has also been shown to elicit phenotypic changes in breast cancer cells. For example, soft matrix with low compression dedifferentiates breast cancer cells towards an estrogen receptor (ER) -negative undifferentiated phenotype and upregulates genes associated with resistance for several chemotherapeutic agents[12–15]. While emerging evidence suggests that softening of the matrix has a major impact on tumor cell identity, little is known how softening of the matrix impacts on the tumor resident immune cells.

Recent findings indicate that primary ex vivo cultures of human tumors in 3D matrices preserve the viability and function of tumor resident immune cells[16–18]. These tumor resident immune cells in ex vivo cultures remain responsive to the anti-PD1 immune checkpoint inhibitors[16,18], and can be used to model compounds that selectively activate and deplete relevant immune cell types[17].

Here we investigated the impact of matrix stiffness on the TIL composition, and activity in patient derived explant cultures (PDEC), which can be grown in specific bioinert matrices tuned for stiffness. We provide evidence that the low compressive forces of a soft matrix do not alter the composition of TILs and their activity directly. Instead, the soft matrix caused these alterations via epithelial tumor cells through triggering upregulation of the cyclooxygenase 2 (COX2) pathway activity and secretion of fibroblast growth factor 2 (FGF2), leading to subsequent immune suppression. We also show that FGF2 expression in breast cancers coincides with low infiltration of T and B-cells and enriched M2-like macrophage gene signature. These studies reveal how the low compression triggers COX2-mediated FGF2 secretion from tumor tissue, promoting local immunosuppression and M2-polarization of macrophages.

## Results

### Tumor resident immune cells are preserved in both soft and stiff NC-gel cultured PDECs

To explore the effect of the mechanical microenvironment on TILs, patient-derived primary tumor tissue was processed into 3D PDECs as previously described[12]. For each sample, we used two concentrations (0.3% and 1.0%) of nanocellulose (NC) matrices (GrowDex from UPM Biomedicals) to prepare "soft" and "stiff" 3D scaffolds for the explants (Fig. 1A). Plant-based NC lacks the matrix-bound growth factors, which are often present in hydrogels used for 3D cultures[19]. Matrigel is a current golden standard matrix in 3D cultures, hence used as a control for bioinert matrices in retraining the immune cell composition. In oscillatory rheology analysis, the average shear storage modulus (G', or stiffness) of the softer NC matrix was 12 Pa, while the more concentrated NC matrix had a stiffness approaching 1 kPa (Fig. 1B, Supplementary Fig. 1A). The stiffness of the Matrigel is approximately 76 Pa[12].

Firstly, the flow cytometric analysis showed that soft NCs and Matrigel were both able to preserve the main immune cell types in 5-day cultures and no statistically significant differences were thus observed in relative T-cell, NKT-cell, NK-cell, or myeloid cell numbers between the uncultured and cultured samples (Fig. 1C). In general, PDEC cultures had relatively 10-20% fewer immune cells than the uncultured samples (Fig. 1D).

The time-lapse imaging after 6 days of ex vivo culture demonstrated motility of TILs in the NC and Matrigel cultures. (Fig. 1E, movies 1–9, Supplementary Fig. 1B). Furthermore, we stained the cultures with immunofluorescent markers for cell death (cleaved caspase 3, CC3) and for proliferation (ki67). Neither consistent apoptosis nor proliferation was observed in any of the conditions (Fig. 1F, Supplementary Fig. 1C).

### PDEC cultures in soft NC lose immune response-specific gene signatures and feature high levels of immunosuppressive cytokines

Next, we explored the pathway activity in PDECs cultured in the soft NC matrix and in the corresponding uncultured samples through transcriptomic analysis. The principal component analysis of transcriptomes showed clustering according to the matrix scaffold across the first three principal components (Fig. 1G). Gene Set Enrichment Analysis (GSEA) with immunologically relevant pathways revealed that soft NC cultures downregulates gene sets associated with active immune response and major histocompatibility complex (MHCII) mediated antigen presentation (Fig. 1H, Supplementary Fig. 2A)[20,21]. Furthermore, the gene set for immunosuppressive cytokine interleukin-4 (IL-4)[22] signaling was upregulated in the soft NC, while the gene set for immune activating cytokine IL-12 signaling[23] was downregulated (Fig. 1H, Supplementary Fig. 2A). These results suggested that soft matrix may create an immunosuppressive microenvironment. To exclude the possibility that the immune suppressive microenvironment was a consequence from the lack of IL-2 in the culture media, in a subset of experiments, we cultured PDECs with matrix-conjugated IL-2 supplementation. However, this did not change the transcriptomic landscape seen in the soft matrix (Supplementary Fig. 2B).

In a parallel experiment, we determined whether matrix stiffness influences immune cells directly or secondary through the tumor tissue in the PDECs. We cultured naïve immune cells from peripheral blood mononuclear cells (PBMCs) alone in stiff and soft NC matrices and compared the cytokine profiles and cell numbers to the corresponding PDEC cultures. A panel of cytokines, that have previously been linked to anticancer immune response were measured[24]. Our panel contained specific antibodies for IL-10, transforming growth factor β (TGF- β), IL-6, IL-1β, IL-2, IL-4, and interferon-γ (IFN-γ). We found that in comparison to the stiff matrix grown PDECs in the soft matrix grown PDECs, the expression levels of immunosuppressive cytokines TGF-β[25] and IL-10[26] were upregulated (Fig. 1I, Supplementary Fig. 3A). Moreover, immune activating cytokine IL-1β[27] was downregulated in soft matrix in all patient samples. Tumor necrosis factor alpha (TNF-α), and IL-6 were not as clearly altered between the soft and the stiff conditions (Fig. 1I, Supplementary Fig. 3A). The expression levels of IL-2, IL-4, and INF-γ were undetectable in all patient samples in all conditions (Supplementary Fig. 3A). As PDECs contain both tumor cells and tumor resident immune cells, we could not identify whether the cytokines were produced by the tumor cells or by tumor resident immune cells.

In the PBMC cultures, IL-10, TGF-γ, IL-6, and IL-1β cytokine levels were not detectable, suggesting that matrix alone without tumor tissue is not sufficient to induce production of these cytokines from immune cells (Fig. 1J, Supplementary Fig. 3B). Moreover, there were no significant differences between the cell numbers of main immune cell

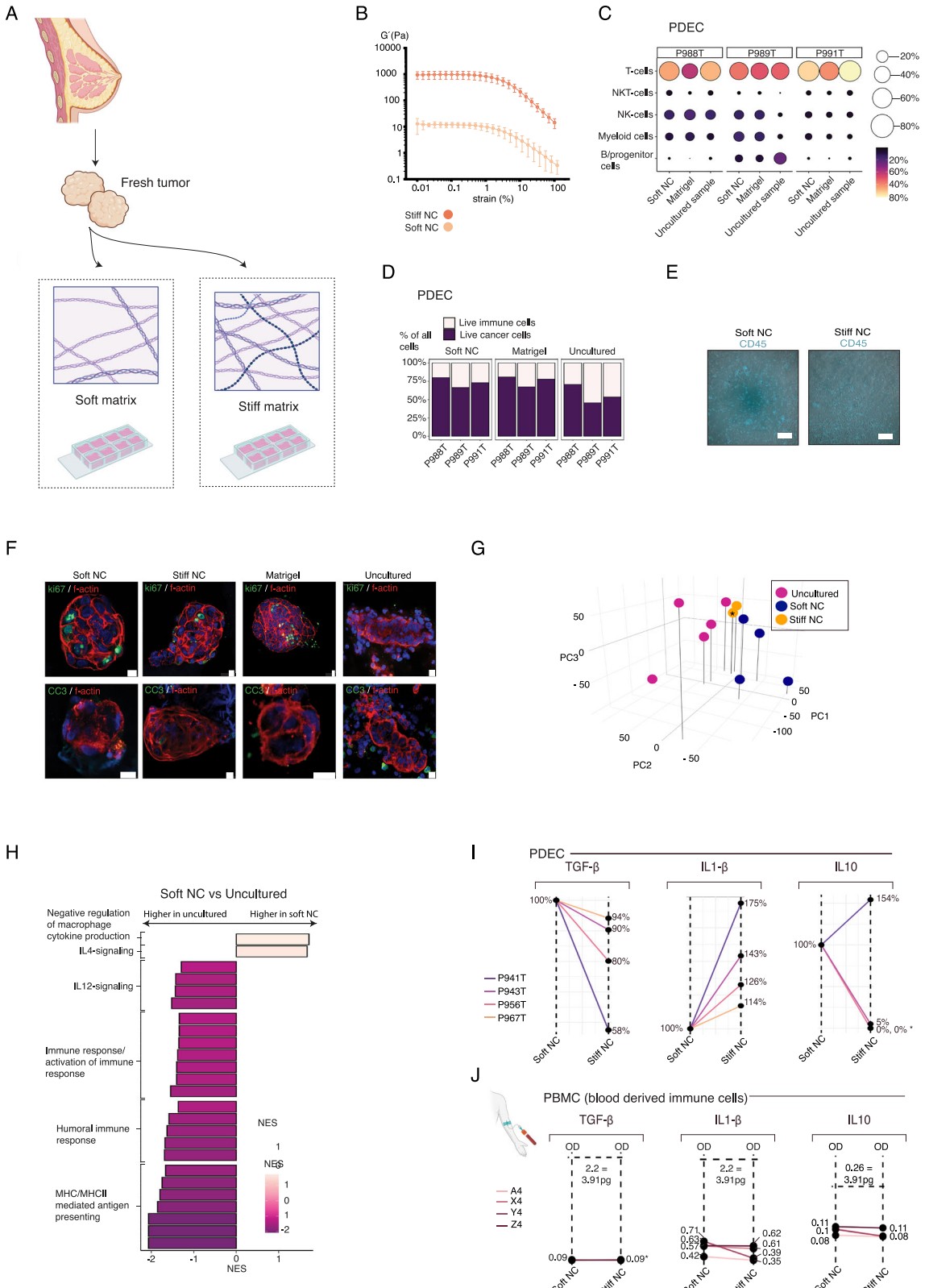

subtypes, between Matrigel and soft and stiff NC cultured PBMCs (Supplementary Fig. 3C). These results indicate that soft matrix triggers production of immunosuppressive cytokines, while downregulating immune activating cytokines via tumor tissue. As fresh isolated PBMCs did not produce observable levels of cytokines in either soft or stiff NC matrix, the presence of tumor tissue was necessary for the matrix effect on cytokine production.

## PDEC cultures in the soft matrix lose cytotoxic CD8 + T- cells and gain M2-like macrophages

For detailed analysis of the immune cell composition, TILs from two fresh uncultured tumors and the corresponding soft NC cultured PDECs were analyzed with a single cell RNA sequencing (scRNASeq). Mast cells, NKT, T-cells, myeloid cells, and B-cells were detected in the soft matrix grown PDECS after 7 days of culture (Fig. 2A–C).

**Fig. 1 | Soft matrix preserves the immune cell content of the original tumor tissue in PDECs. A** Schematic representation of the patient derived explant cultures (PDEC) in soft and stiff microenvironments. Created in BioRender. Peura, A. (2025). https://BioRender.com/q2r8igh. **B** The rheological strain sweep measurements of the soft and stiff nanocellulose (NC) matrices (*n* = 5 per matrix stiffness). Dots represent the medium and error bars represent the standard deviation. **C** The relative percentages of main tumor infiltrated leukocytes (TIL) subtypes from three patient samples cultured in soft nanocellulose (NC) gel and Matrigel in comparison to fresh uncultured tumor, the size of the dot and the color bar indicate the percentage of the cell type in the sample. **D** The relative numbers of TIL and tumor cells in PDEC samples (*n* = 3). **E** PDEC samples cultured for 7 days and stained with CD45. The corresponding live video is in the Supplementary Movie 1-9. The scale bar

represents 100 µm. Two tumor samples and one normal sample was grown in each matrix. **F** PDEC samples cultured for 7 days and stained with cleaved caspase 3 (CC3) and Ki67 (*n* = 3). The scale bar represents 10 µm. **G** Principal component analysis (PCA) of the bulk-mRNA from PDEC samples (*n* = 5 for the uncultured, *n* = 4 for soft NC). The star indicates that two stiff-NC samples are overlapping in the figure. **H** Gene Set Enrichment Analysis (GSEA) analysis of gene sets upregulated in the soft NC in comparison to the uncultured sample. **I** ELISA cytokine assay of four individual patients cultured as PDECs for 3 days. The star indicates that several donors are overlapping in the plot. **J** ELISA cytokine analysis of four individual peripheral blood mononuclear cell (PBMC) donors with similar cell numbers per matrix and dilution factors as detected from PDEC samples. Levels for all cytokines were below the detection level and hence values are represented as OD-values.

Interestingly, we found that while the total proportion of T-cells was similar between the uncultured samples (44%) and the soft NC matrix (46%), the proportion of the cytotoxic effector memory CD8 + T-cells sharply decreased from 10% level in uncultured samples to 0.5% in the soft matrix (Fig. 2D, E). The memory CD8 + T-cells comprise a subset of T-cells that have responded to cognate antigen and persist long-term[28]. In contrast to the striking loss of CD8+ effector memory CD8 + T-cells, the proportion of CD4 + T-helper cells nearly doubled from 9% in the uncultured samples to 20% in the soft matrix (Fig. 2D, E). To confirm that the observed loss of CD8 + T-cells was due to the reduced stiffness of the microenvironment, we next compared three individual patient samples in soft and stiff gel PDEC-cultures. Instead of soft and stiff NC, we needed to use another bioinert hydrogel, PeptiGel (PG), because the stiff NC gel was observed to lose its stiffness in the long-term cultures as it diffused into the culture medium (Supplementary Fig. 3D). Also, the flow cytometrical analyses were challenging due to the difficulties dissociating NC matrix. The ability of PG gels to maintain their stiffness in the long-term ex vivo cultures was likely due to stronger crosslinking by the culture medium components than that occurring in NC gels[29]. The soft PG matrix had a stiffness of approximately 150 Pa, while the stiffness of stiff PG matrix was approximately 6.5 kPa (Supplementary Fig. 3E). Soft and stiff PeptiGel cultures maintained approximately similar numbers of T-cells, B-cells, APC and NK/NKT cells alive after 3 days of culture (Supplementary Fig. 3F). The patient samples were embedded into soft and stiff PG matrix and the CD8 + T-cells were analyzed with flow cytometry (Fig. 2F, G). We were able to verify, that the total number of CD8 + T-cells, as well as the number of effector memory, and TEMRA subtypes were significantly diminished in the soft PG matrix in comparison to the stiff matrix (Fig. 2F, G).

In addition to the observed T-cell subtype changes, we found that the proportions of mast cells and B-cells were markedly lower in the soft matrix cultured PDECs as compared to the uncultured samples from which they originated from. The level of the mast cells diminished from 13.7% to 4.75% and the B-cells from 9.3% to 1.8% (Fig. 2D, E). A slight soft matrix related increase was observed also in the myeloid cell population (Fig. 2D, E). About 16-18% of the cells in the data remained unassigned as these cells represented some markers of T-cells, such as CD2+ but also had a low expression of CD3+ (Fig. 2D, E, Supplementary Fig. 4A). Markers used for cluster identification are shown in Supplementary 4A, B.

We further investigated the macrophage phenotypes in scRNASeq data through defining differentially expressed genes between myeloid cells present in the uncultured samples and soft NC cultures (Fig. 2H). Fast Gene Set Enrichment Analysis (Fgsea) with several M1 and M2 macrophage related gene sets revealed the diminished M1-like macrophage phenotype compared to M2-like phenotype in the soft PDEC-NC cultures (Fig. 2I). The list of gene sets and associated source publications are provided in source data file. In the scRNASeq analysis, we used uncultured sample as a control for the soft NC cultures. To further explore macrophage phenotypes in stiff versus soft PDEC

cultures, we cultured tumors from 8 individual patients in stiff and soft PG matrices and analyzed the median expression of two M2 markers, CD206 and CD163, in the macrophage population of the PDECs (Fig. 2J). In 7 samples, both CD163 and CD206 expressing cell populations were higher in the soft PG matrix compared to the stiff PG matrix (*p* = 0.04 for CD206 and 0.043 for CD163 (Fig. 2J). One sample (P1825T) did not follow this trend for reasons that are unclear.

In summary, PDECs cultured in the soft matrix present an immunosuppressive cytokine profile, downregulation of many immune activation specific gene sets, loss of effector memory CD8 + T and simultaneous enrichment of CD4 + T-cells, and finally polarization of the macrophage phenotype from M1 to M2 subtype. Since similar changes were not observed in the naïve matrix-embedded immune cells, we concluded that the immunosuppressive effects in the soft matrix cultures originated from the epithelial tumor tissue component present in the PDEC-cultures.

## FGF2 and COX2 pathways contribute to the immune suppressive microenvironment in the soft matrix cultures

To explore how soft matrix stiffness affects the tumor tissue in PDECs, we defined the top differentially expressed pathways between PDECs grown in the soft NC in comparison to the original tumor. The GSEA analysis of cell signaling and cancer-associated pathways further revealed that the soft NC cultured PDECs upregulated genes and gene expression profiles of fibroblast growth factor signaling (FGF), epithelial mesenchymal transition (EMT), mesenchymal breast cancer related pathways as well as mammary stem cell/luminal progenitor like phenotype e.g., Wingless/Integrated signaling (WNT), and TGF-β pathways (Fig. 3A, B, Supplementary Figs. 5A, B, 6A, and Source Data)[30–33]. In addition, H3K27 DNA methylation signature was upregulated in the soft NC (Supplementary Fig. 6B). We have previously shown that the soft matrix associated enhancer of zeste homolog 2 (EZH2) upregulation drives breast cancer cells towards more undifferentiated cell state[12,34].

Besides gene sets indicating the mesenchymal phenotype of the tumor tissue, gene sets indicating FGF and FGF receptor (FGFR) signaling were among the most enriched gene sets in the soft NC (Fig. 3A). We confirmed the upregulation of FGF2 in the soft NC cultures with western blot analysis. The paracrine low-molecular weight (lmw) 18 kDa form of FGF2 was upregulated in protein level both in the soft NC PDEC cultures vs the original uncultured tumor and in the soft PG vs stiff PG PDEC cultures (Fig. 3C–E)[35]. Earlier studies have shown that within the tumor microenvironment, lmw-FGF2 promotes the M2-phenotype of tumor-associated macrophages and the exhaustion of CD8 + T-cells[36,37]. Therefore, we considered FGF2 as a candidate growth factor contributing to the immunosuppressive features seen in the soft matrix.

Another pathway of interest that was enriched in the soft matrix grown PDECs was COX2 pathway since earlier studies have linked the COX2-prostaglandin E2 (PGE2) pathway to immune suppression in the tumor immune microenvironment (TIME)[38] and to FGF2 expression in

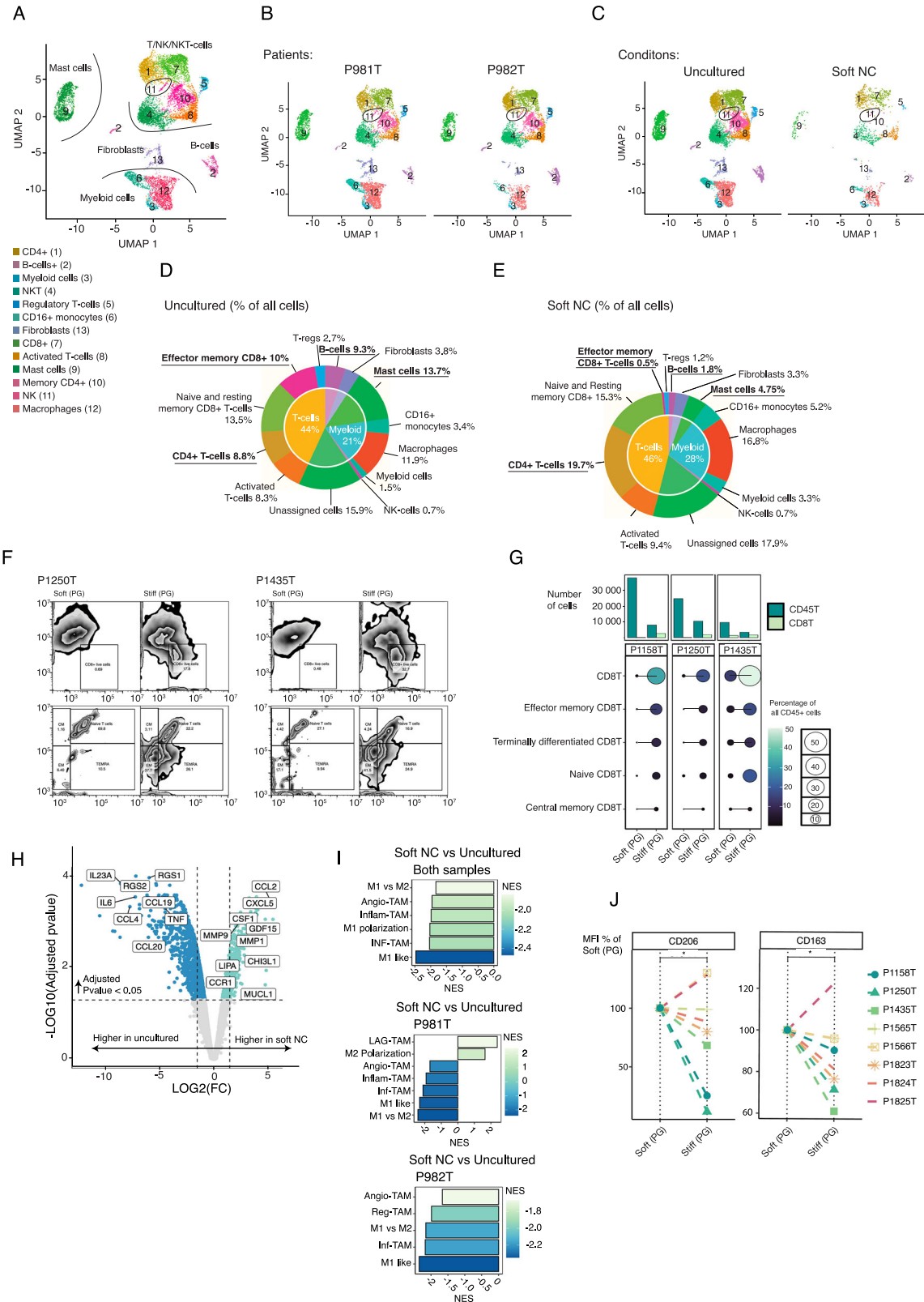

endometrial and pancreatic cancer[39,40]. We found that the mRNA for Prostaglandin E Synthase (PTGES), which is a key enzyme in the production of PGE2 and prostaglandin-endoperoxide synthase 2 (PTGS2), a gene for COX2 pathway, were significantly up-regulated in the soft PDEC-NC cultures as compared to corresponding uncultured samples (Fig. 3B, C). In addition, western blot analysis of samples from 3 patients confirmed the increased COX2 protein expression in both the soft NC samples vs original uncultured samples and in soft PG samples vs stiff PG cultured PDEC samples (Fig. 3D, E).

Furthermore, among the other most differentially expressed genes between the soft and the uncultured samples were several immunosuppressive cytokines such as IL-33, IL-11, as well as HS3ST3B1, a gene associated with the production of heparin and heparan sulfate[41,42].

**Fig. 2 | CD8 + T-cells are lost, and macrophages are polarized towards M2-like phenotype in the soft matrix PDEC cultures. A** Uniform Manifold Approximation and Projection (UMAP)-representation of all cell types identified from the scRNA-seq data. Patient samples were cultured in soft nanocellulose (NC) matrix for 7 days and uncultured original tumor sample was used as a control ($n = 2$). **B** UMAP representation of individual patient samples and (**C**), for culture conditions. **D**, **E** Relative amount of main cell types (inner circle) and all cell types (outer circle) in the uncultured sample and sample cultured in the soft NC gel. Clusters that represented more than 5% of cells in the uncultured sample and lost or gained more than 50% of their cells are underlined and bolded. **F** The flow cytometry gating strategy for CD8 + T-cells of two patient samples grown as patient derived explant cultures (PDEC) in soft and stiff PeptiGel (PG) gels. **G** Quantification of main CD8 + T-cell subtypes in three patient samples grown in soft and stiff PG gels. Significant comparisons are indicated with line ($n = 3$). Statistic was calculated with one-sided t-test, exact p-values are shown in the source data. **H** Top differentially expressed genes in pseudobulk simulated data from myeloid cell/macrophage cluster (clusters macrophages, CD16+ myeloid cells, and myeloid cells) between the uncultured sample and sample cultured in the soft NC gels. Statistics was calculated by EdgeR. **I** Fgsea-analysis with macrophage related gene sets (source data file) from pseudobulk-data, statistics was calculated with. All represented gene sets are significant with padj < 0.05. **J** Median fluorescence intensity of two M2-macrophage related markers, CD206 and CD163, between the soft and stiff PG gel cultures ($n = 8$). Statistics calculated with one-sided Wilcox ranked sum test. Exact *p*-values are shown in the source data.

We investigated further whether *FGF2* and *PTGS2* are stiffness regulated genes outside our own mRNA sequencing through exploring the expression level of these genes in PDECs cultured either in ultra-stiff bioinert matrix (Agarose-based LMx-Ag; stiffness G′ = 46 kPa +/-8 kPa) or soft bioinert matrix (Egg white based LMx-Ew; stiffness G′ = 0.04 ± 0.03 kPa)[12]. The corresponding uncultured fresh tumor tissues were included in the analysis as a representative of the stiffest condition. We found that the expressions of *PTGS2, PTGES*, and *FGF2* were clearly higher in the PDECs grown in soft matrix than in the PDECs grown in stiff matrix or in the uncultured samples (Fig. 3F). Other potentially stiffness regulated genes were *IL-33, HS3ST3B1*, and *IL-11*, which were among the most differentially expressed genes between the PDECs grown in the soft matrix compared to the original uncultured stiff tumor (Fig. 3B, F). These top differentially expressed genes, *IL-33, HS3ST3B1, IL-11, FGF2*, and *PTGES* were observed unchanged in our scRNASeq data, which included only the PDEC resident TILs (Fig. 3G, Supplementary Fig. 6C). These results suggest that PGE2 and FGF2 are stiffness regulated genes originating from the epithelial tumor cells in PDECs (Fig. 1G).

### FGF2-COX2 pathway promotes M2-like macrophage phenotype

We next investigated whether FGF2, produced by the tumor cells in the soft matrix PDEC cultures, was able to contribute to the macrophage polarization from M1 to immunosuppressive M2 type. Naïve PBMC derived immune cells were first differentiated into macrophages by selecting CD14+ cells and culturing them for 5 days with the macrophage colony stimulating factor (M-CSF) (Fig. 4A). Heparin is an important cofactor for FGF2-signaling as it stabilizes extracellular FGF2[43]. Macrophages were treated with IFNγ and lipopolysaccharide (LPS) to polarize them into M1 and for M2 polarization, the M0 macrophages were treated with IL4 + IL13[44]. As expected, INFγ + LPS decreased the CD163 and CD206 expression, consistent with polarization to M1. Furthermore, IL4 + IL13 treatment decreased CD163, while increasing expression of CD206-positive cells. The resulting CD163low; CD206high phenotype is consistent with M2a subtype of M2 macrophages[45–47]. Intriguingly, FGF2 + heparin increased the CD163 and lowered the CD206 expression, resembling M2c macrophages characterized by a CD163high/CD206low phenotype (Fig. 4B, Supplementary Fig. 6D). M2c macrophages mediate phagocytosis, immunosuppression, angiogenesis, and the development of tissue fibrosis, and they are polarized with IL10, TGFβ, and glucocorticoids[45,48]. Moreover, to further characterize the FGF2-induced macrophage phenotype, we performed a qRT-PCR analysis for macrophages treated with FGF2, heparin, or both with more markers to macrophage phenotype (Fig. 4C, Supplementary Fig. 6E). We were able to conclude, that while control sample had a higher expression of M1-related marker CXCL9, the FGF2 + heparin treatment induced especially the expression of M2-related CCL17 (Fig. 4C).

As FGF2 induced macrophage polarization appeared to be dependent on heparin as a cofactor, we further investigated whether heparin or heparan sulfate production was increased in the soft matrix. The heparin production was clearly higher in the soft NC compared to the stiff NC (Fig. 4D) in tumors cultured for 3 days as PDECs.

The COX-2 (cyclooxygenase-2) pathway and FGF2 (Fibroblast Growth Factor 2) are interconnected through several mechanisms, particularly in the context of inflammation and cancer[39,40,49]. Next, we explored whether COX2 pathway mediated the FGF2 expression in PDECs. We treated PDECs with Ketoprofen, a COX1/2 specific inhibitor[50], and Celecoxib, which is a COX2 specific inhibitor[51] and observed a significant downregulation of FGF2 (Fig. 4E, F). To address whether COX2-FGF2 pathway depolarizes macrophages we treated soft matrix cultured PDECs with NSC12, which chelates FGFs[52], alone or in combination with Celecoxib. While FGF2 inhibition with NSC12 downregulated the expression of CD163, it did not significantly affect CD206 (Fig. 4G). However, the combination of NSC12 and Celecoxib suppressed both CD163 and CD206 suggesting the depolarization of several M2 subtypes (Fig. 4G).

### FGF2 expression corresponds to the immune suppressive microenvironment in the primary breast cancer samples

To explore the FGF2 expression pattern in estrogen receptor positive (ER + ) and aggressive triple negative breast cancer (TNBC) subtypes, we performed immunohistochemical staining for FGF2 in 38 ER+ and 18 TNBC primary tumor samples (Fig. 5A, Supplementary Fig. 7). We observed that about one-fifth of ER+ tumor samples were positive for FGF2 expression both epithelial and mesenchymal (Supplementary Fig. 7), whereas over 80% of TNBC samples were positive for FGF2 (Fig. 5A).

To define the spatial pattern of FGF2 expression in relation to the immune cell infiltrates, we performed a multiplex IHC staining with a panel of four general immune cell markers (CD68, CD20, CD3, CD45, and pan cytokeratin) for 11 ER+ primary breast cancer samples selected from the previously stained samples (Fig. 5B–F, Supplementary Fig. 12). These samples were first divided into FGF2 positive and FGF2 negative subgroups. The location of TIL-rich areas (CD45 + ) within epithelial tumor tissue (EPCAM +) was defined through visual inspection (Fig. 5B), and the whole tissue immune cell content was quantitated through image analysis (Fig. 5C, G), blind to the FGF2 categories. FGF2 negative tumors contained significantly more T- (CD3 + ) and B-cells (CD20 + ), than FGF2 positive tumors (Fig. 5E, F, Supplementary Fig. 12). The number of macrophages (CD68) and all TILs (CD45) were similar irrespective of the FGF2 status in the tumor samples (Fig. 5C, D).

To evaluate the presence of M2-macrophages in FGF-signaling enriched areas, we quantified the spatial mRNA expression of *FGF2*-signaling in 18 TNBC samples derived from 7 individual patients (Fig. 6A, Supplementary Figs. 8, 9). To specifically evaluate tumor cell associated FGF2 signaling, we analyzed spots containing over 75% of tumor cell -derived reads, following the method published in Bassiouni et al.[53]. We found in ssGSEA analysis that specific areas enriched in FGF2 signaling associated with M2-macrophages signatures.

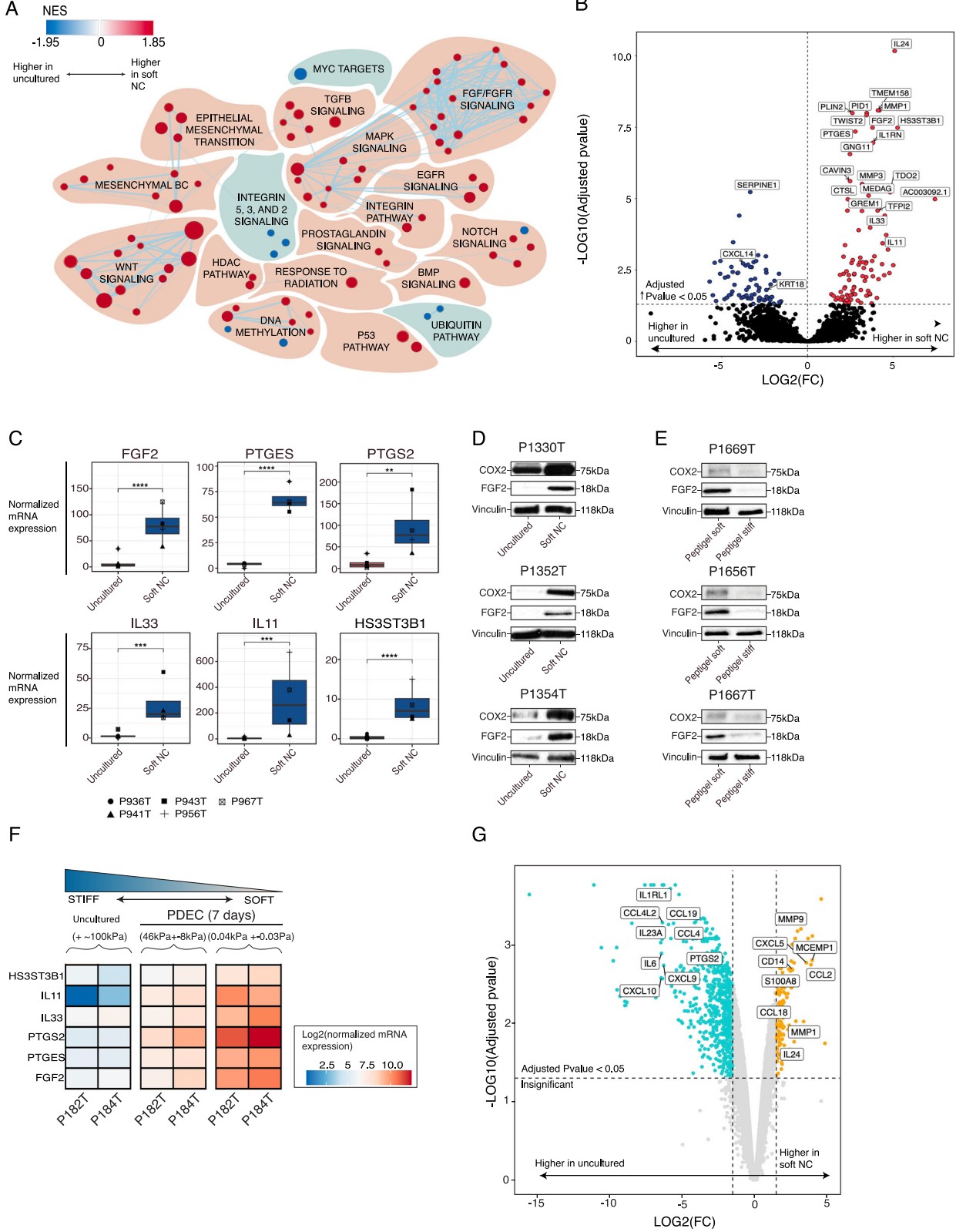

Moreover, the signatures for T-, B-, and CD8 + T-cells were diminished in FGF2-rich areas (Fig. 6A, B, Supplementary Figs. 8, 9).

These analyses couple high spatial *FGF* expression status and FGF- signaling activity with high macrophage content and specifically enrichment of M2-macrophage signatures. Moreover, FGF2 enriched areas had less T- and B-cell infiltration, especially less

CD8 + T-cells. These changes in immune cell subtypes in FGF2-rich areas, namely enrichment of M2-macrophages and loss of CD8 + T- and B-cells, resembled the immune cell composition of soft NC cultured PDECs (Fig. 2D, E). These findings further suggest a role for FGF2 in the local immunosuppression of human primary breast cancers.

**Fig. 3 | Upregulation of FGF2 and COX2 expression in the soft matrix PDEC cultures. A** Top differentially upregulated (red) and downregulated (blue) gene sets between the soft nanocellulose (NC) and the original uncultured tumor. The size of the dot represents the gene set size. The list of genes in the gene sets is shown in source data. Only significant (FDR < 0.2) sets are visualized. **B** Top differentially expressed genes between soft NC and the original uncultured sample. Statistics was calculated by DEseq2. **C** mRNA level of several immunity and fibroblast growth factor 2 cyclooxygenase 2 (FGF2-COX2) related genes between soft and original uncultured tumor ($n = 5$). Statistics was calculated with DEseq2. Exact

adjusted p-values are 3.2649E-08 for FGF2, 4.4E-08 for PTGES, 0.0011 for PTGS2, 0.00037 for IL11, 0.00010 for IL33 and 3.26E-08 for HS3ST3B1. **D, E** Protein level analysis of COX2 and FGF2 in three different patients between the soft and the original uncultured tumor, as well as between the soft and stiff PeptiGel (PG). $N = 3$ for PeptiGels and $n = 3$ for soft GrowDex. **F** Normalized mRNA sequencing data shows the expression of selected genes across the matrices with different stiffnesses. **G** Top differentially expressed genes in pseudobulk simulated expression data between immune cells cultured in the soft NC and in the uncultured samples ($n = 2$). Statistics was calculated by EdgeR.

## Discussion

In this work, we first demonstrate how a sudden softening of matrix stiffness affects tumor-resident immune cells in PDEC cultures, shifting the immune landscape towards an immunosuppressive state characterized by the loss of CD8 + T-cells and increased M2 macrophage polarization. We then show that these changes are driven by enhanced COX-FGF2 signaling, which arises from the dedifferentiation of tumor cells in response to the sudden softening of their environment.

Many studies have established that tumors can be devoid of active immune cells (cold tumors) or contain active immune cells (hot tumors), which may predict the tumors responses to immune checkpoint inhibitors[54,55]. Hallmarks of immune suppressed tumors include the presence of immunosuppressive cytokines such as IL-6, IL-10, TGF-β, and tumor-promoting immune cell phenotypes such as regulatory T-cells and M2-like macrophages[56]. In the present study we show that PDEC-cultures maintain similar immune cell composition as in the original primary tumor. The number of main immune cell subtypes (T-, NK-, B-, and myeloid cells) was similar in the PDEC-cultures compared to the uncultured tumor. We compared the activities of immune cells in stiff and soft matrix grown PDECs and noted specifically in the soft matrix an elevated secretion of immune suppressive cytokines (TGF-β, IL-10, IL-4) with concomitant downregulation of immune activating cytokines IL-12 and IL-1β. Moreover, in the soft matrix grown PDECs, the macrophage phenotype was polarized towards more M2-like and the number of effector memory CD8 + T-cells was diminished. These findings suggest that soft matrix promotes the formation of an immunosuppressive tumor immune microenvironment in the PDEC cultures.

Since the formation of immunosuppressive tumor microenvironment occurred only in PDECs, and not in the pure immune cells containing PBMC-cultures, we conclude that the observed immune suppressive phenotype in the soft matrix cultures was dependent on the presence of tumor cells. Our earlier studies have shown that tumor cells alongside the immune cells are the predominant cell types in PDECs, since the collagenase treatment removes adipose and connective tissues[12]. Transcriptomic studies of PDECs grown in the soft matrix revealed gene sets corresponding to partial EMT as well as upregulation of mesenchymal TNBC signatures, and signaling pathways associated with mammary stem cell/luminal progenitor like phenotype (Notch, TGF-β, WNT)[30–33]. These results suggest that the soft matrix induces loss of original differentiated luminal breast cancer phenotype and promotes dedifferentiation towards poorly differentiated mesenchymal and luminal progenitor phenotype. The observed dedifferentiated phenotype of soft matrix grown PDECs is consistent with our earlier studies that have established a critical role for stiff matrix in maintaining the original differentiated and ER+ phenotype[12,13]. Mechanistically, the increased matrix stiffness signals via inactivation of transcription regulator EZH2 resulting in decreased H3K27 methylation and expression of differentiation related genes such as estrogen receptor[12]. We observed enrichment of H3K27 methylation gene signatures in the soft matrix grown PDECs, which further supports the dedifferentiation of epithelial tumor tissue in the soft matrix.

We observed that the soft matrix dependent dedifferentiation of tumor tissue led to upregulation of IL-10 and TGF-β, which are well known mediators of immunosuppression[26,57]. Additionally, the soft matrix cultured PDECs also upregulated COX2-FGF2 pathway. Both paracrine factors of COX2-FGF2 pathway, PGE2, and FGF2 have been found to enhance M2 phenotype in mouse models of breast and lung cancer[36,58]. We show that FGF2 + heparin treatment increased CD163 and lowered the CD206 expression in M0 macrophages. While the phenotype of FGF2 + heparin differentiated macrophages requires more thorough characterization in further studies, the appearance of the CD163high phenotype together with secretion of immunosuppressive cytokines IL-10 and TGF-β resemble M2c macrophages characterized by a CD163high/CD206low phenotype also known as glucocorticoids (GC) activated M2 macrophages, M(GC)[45,48,59]. In PDECs, depletion of FGF2 with NCS12 diminished the population of CD163+ macrophages, which demonstrates the requirement of FGF2 for this M2 subtype. The presence of CD163+ macrophages in breast tumor samples has been associated with two times higher risk for disease progression and shorter overall survival[57]. In addition, experiments in mice models have suggested the important role of CD163+ macrophages in tumor growth and metastasis[58]. We show that the tumor cell phenotype is plastic and sudden softening in the tumor microenvironment may alter the tumor cell transcriptome, affecting the factors that cells secrete. These changes further influence tumor-resident immune cells, leading to an enrichment of CD163 + M2-macrophages, which are associated with poor prognosis in breast cancer[59].

In this study, we also present evidence for the role of FGF2 in local immunosuppression through demonstrating absence of T-cells and B-cells in FGF2-rich regions in ER+ primary breast cancers. The total level of macrophages (CD68 + ) was similar in both FGF2 high and FGF2 low tumor sites. We further confirmed the presence of M2 macrophage signature, as well as depletion of CD8 + T-cells from FGF-rich areas from the spatial transcriptomics data.

Contrary to the common belief that tumor is uniformly stiff in comparison to normal breast tissue, recent findings suggest that the progression of breast tumors from the premalignant stage to malignant tumors is associated not only with increased stiffness but also with greater heterogeneity in intratumoral stiffness[60]. In addition, advanced tumors have local soft areas, created via hypoxia or chemotherapy-induced necrosis. Furthermore, the metastatic lesions in the lungs can be substantially softer than the primary tumor[60]. Consistent with our findings suggesting a role for soft matrix in local immunosuppression, previous research has shown that especially in the breast tumor lung metastasis, the microenvironment in the invasive front is more immunosuppressed than the primary tumor, as indicated by enhanced expression of IL-10 and TGF-β, as well as enrichment of M2 macrophages[61].

In summary, in this work we show evidence indicating that soft matrix promotes through tumor epithelial dedifferentiation the activation of COX2-FGF2 pathway leading to immunosuppressive microenvironment in the PDECs. In the soft matrix PDEC-cultures, immunosuppressive cytokines are predominantly present, and tissue macrophages are polarized towards M2-phenotype, specifically

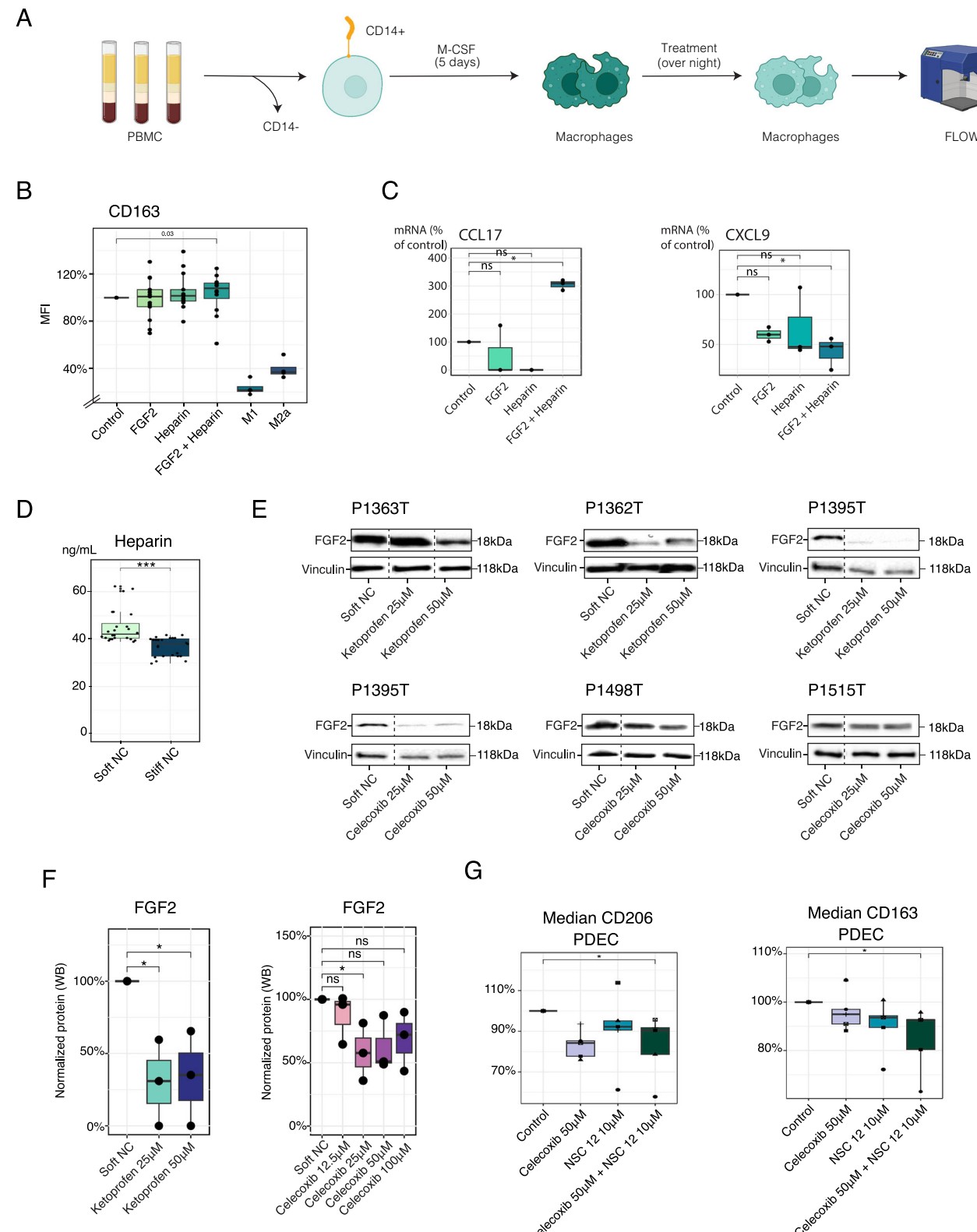

enriching in M(Gc)/CD163+ subtype. There are currently several therapeutic approaches to depolarize M2 macrophages as an immunotherapeutic modality, including e.g. STING-agonists, cytokines and ligands for macrophage scavenger receptors[62]. In addition to increasing understanding of the spatially acting immunosuppressive mechanisms in breast cancer, we believe that the current findings may also open new avenues for therapeutic targeting of FGF2-COX2 pathway as a mean to depolarize M(Gc)/CD163+ macrophages towards

more M1-like, especially in treatment of metastatic breast cancers and primary TNBCs.

## Methods
### Patient Derived Explant Cultures (PDEC)
Primary breast tumor tissue was obtained from elective breast cancer surgeries from Helsinki University Hospital under ethical permit (243/13/03/02/2013/ TMK02157 and HUS/2697/2019 approved by the

**Fig. 4 | FGF2 treatment polarizes macrophages to M(Gc)-like. A** Schematic representation of the isolation of CD14+ cells and differentiation into macrophages. Created in BioRender. Peura, A. (2025). https://BioRender.com/e9a70zq. **B** Median fluorescent intensity of CD163 in the M0 macrophages cultured with fibroblast growth factor 2 (FGF2), heparin + FGF2, heparin overnight and compared with interferon-γ (IFN-γ) + lipopolysaccharide (LPS) (M1) and interleukin (IL-4) + IL-13 (M2a) treated macrophages (M1/M2 $n = 4$, $n = 12$). Statistics was calculated with two-sided wilcox test, exact p-values are shown in the source data. **C** Expression of CCL17, CXCL9, and Tumor necrosis factor alpha (TNF-α) at the mRNA level in the macrophages cultured with FGF2, heparin or with both FGF2 and heparin overnight ($n = 4$). Statistics was calculated with anova and Tukey HSD. Adjusted p-values are 0.031 for CXCL9 and 0.003 for CCL17. **D** Quantification of heparin levels from four patient derived explant cultures (PDEC) samples cultured in the soft and the stiff nanocellulose (NC) matrix for 3 days ($n = 4$). Statistics was calculated with two-sided t-test, exact p-value is 4.22E-07. **E, F** Western blot protein analysis of cyclooxygenase 2 (COX2) and FGF2 from six different patient samples treated with Ketoprofen or Celecoxib. The protein quantitation is represented as boxplot. Statistics was calculated with two-sided t-test, p-values are 0.02 for soft vs Ketoprofen 0.25 ul, 0.03 for soft vs Ketoprofen 50ul and 0.025 for soft vs Celecoxib. **G** Median expression of CD163 and CD206 in macrophages from PDECs with NCS12 or Celecoxib supplementation or with combination ($n = 4$). Statistics was calculated with one-sided t-test, exact p-values are 0.0342 for CD163 and 0,0337 for CD206.

Helsinki University Hospital Ethical Committee). Patients participated in the study by signing an informed consent form following the Declaration of Helsinki principles.

The fresh breast tumor tissue was dissociated into small tumor fragments with collagenase A (Sigma) treatment with gentle shaking (130 rpm) during overnight at +37 °C. In the following day, the dissociated breast tumor tissue was embedded into 3D matrix on an 8- or 18 well chamber slide (Thermo Scientific, Ibidi respectively) and supplemented with 40 μL of matrix and 130-500 μL of MammoCult medium (StemCell technologies, #05620).

MammoCult growth media contained: MammoCult proliferation supplement (StemCell technologies #05622), 10,000 U/mL penicillin/ streptomycin (Lonza), 4 μg/mL heparin, 20 μg/mL gentamicin (Sigma), and 0.1 μg/mL amphotericin B (Biowest). 3D-matrices used in this study were the following: GrowDex (stock 1.5%) (UPM Biomedicals) with concentrations 0.3% and 1.0% diluted to cell culture medium according to manufacturer's instructions, PeptiGel® Alpha 1™ (CELL guidance systems) with concentrations 100% and 20%, and undiluted Matrigel (growth factor reduced, Corning). Medium for PDECs was changed every 48–72 h. Cells were grown in a humidified incubator at +37 °C under 5% CO₂ and atmospheric oxygen levels.

GrowDex-A (UPM Biomedicals) was functionalized with human recombinant biotinylated IL-2 (2 ng/mL) (BT202-025/CF, R&D Systems). 50 μL of IL-2 diluted in media was mixed with 300 μL of GrowDex-A and incubated for 1 h at RT. Functionalized matrix was adjusted to the correct concentration (0.3%) by adding 550 μL of culture media. 1 μM biotin (Sigma, B4639-1G) was added to the growth media to block the free biotin sites.

## Human peripheral blood mononuclear cell (PBMC)-cultures and related assays

Buffy coat was obtained from Finnish Red Cross blood service under ethical permit 15/2023. Ficoll-Paque density centrifugation was used for isolation of PBMCs according to the standard protocol. After isolation, the cells were stored at −140 °C.

For the cytokine profiling, the cells were thawed and placed into a 3D-matrix together with RPMI or IDMD-medium containing 10% of heat-inactivated fetal bovine serum (FBS) (Serana), and 10,000 U/mL penicillin/streptomycin (Lonza). On day 3, the medium was collected, and cytokines were analyzed as described in cytokine analysis.

For the macrophage differentiation assay, PBMCs were thawed and CD14+ monocytes were isolated with following protocol modified from the manufacture's protocol. First, tubes containing 3–5*10^7 fresh isolated PBMCs were thawed in a warm water and diluted into a staining buffer containing phosphate buffered saline solution (PBS), 2 mM ethylenediamine tetra acetic acid (EDTA) (Cat: E9884, Sigma-Aldrich), and 1% heat-inactivated FBS (Serana). Next, the cells were centrifuged with 400 g for 5 min and resuspended into 200 μL of the staining buffer. 10 μL of CD14 MicroBeads (Miltenyi Biotec) per 10^7 cells were added in the solution, the cells were vortexed followed by incubation for 7 min together with beads at +4 °C. After the incubation, the cells were vortexed again and incubated for 7 min at +4 °C. The remaining beads were washed off by diluting cell-bead solution with 1 mL of staining buffer and centrifuging the samples at 400 g for 5 min. Next, the samples were diluted into 450 μL staining buffer. MACS-columns (Miltenyi Biotec, 130-042-401) were used for separating the CD14+ cells from the remaining PBMCs. The columns were placed into MACS®- separator and washed once with staining buffer. The CD14-beads stained PBMC-solution was added to the columns and washed 3 times with the staining buffer. Finally, the CD14+ cells were forced out from the column by firmly pushing them with a syringe. After this, the cells were placed into a 96-well plate (100,000 cells per well) with a growth medium containing Iscove's Modified Dulbecco's Medium (IMDM) (Gibgo), heat-inactivated fetal bovine serum (FBS) (Serana), penicillin-streptomycin (Lonza) and 50 ng/mL macrophage-colony stimulating factor (M-CSF) (Miltenyi Biotec). The cells were cultured for 5 days during which the medium was changed once. On day 5, the cells were treated with 50 ng/mL of heparin and/or 100 ng/mL of recombinant human FGF2 (R&D Systems) and/or 20 ng/mL IFN-γ (Sigma-Aldrich no. I3265) and 100 ng/mL lipopolysaccharide (LPS; Sigma-Aldrich no. L2630, *E. coli* 0111:B4 strain) or recombinant IL-4 and IL-13 and incubated for 24 h.

Afterwards, the cells were either detached with 400 μL of macrophage detachment solution (40 min at +4 °C and 20 min at RT) or processed for qRT-PCR analysis as described in the qRT-PCR section. Next, the macrophages were stained with antibodies as described in flow cytometry section and described in Supplementary Fig. 11 and Supplementary table 1 and relative number of M1 and M2 macrophages were calculated.

## Rheology

Rheological characterization of NC- (0.3% and 1%) and PeptiGels (20% and 100%) were carried out using MCR 302 rheometer (Anton Paar) with a 25 mm parallel-plate geometry. After sample loading, pre-shearing was performed at $1 s^{-1}$ for 30 s, followed by 5 min rest to minimize the effect of sample shear history. Subsequently, dynamic oscillatory strain sweep was conducted in the range of 0.01–100% strain at a frequency of 1 Hz. The nanocelluloses and the 20% PeptiGel were measured directly after sample dilution, but the 100% PeptiGel was first conditioned with medium. In brief, the 100% PeptiGel was pipetted onto rheometer's Peltier plate using a circular mold 25 mm in diameter. To induce crosslinking, the gel was covered with Mammocult medium and incubated at 37 °C for 30 min. Excess medium was removed before the measurements were started. The nanocelluloses were measured at a temperature of 20 °C and the PeptiGels at 37 °C. The viscoelastic properties of the samples were defined by the shear storage modulus (G′, representing matrix stiffness), the shear loss modulus (G″, representing viscous behaviour), and the phase angle (δ, related to the ratio of G″ and G′). The average G′, G″ and δ were calculated from the linear viscoelastic region at 0.1% strain.

## Immunofluorescent staining (IF)

IF was used to determine the cell viability in 3D cultures. The PDEC-samples were stained with the immunofluorescence markers for cell

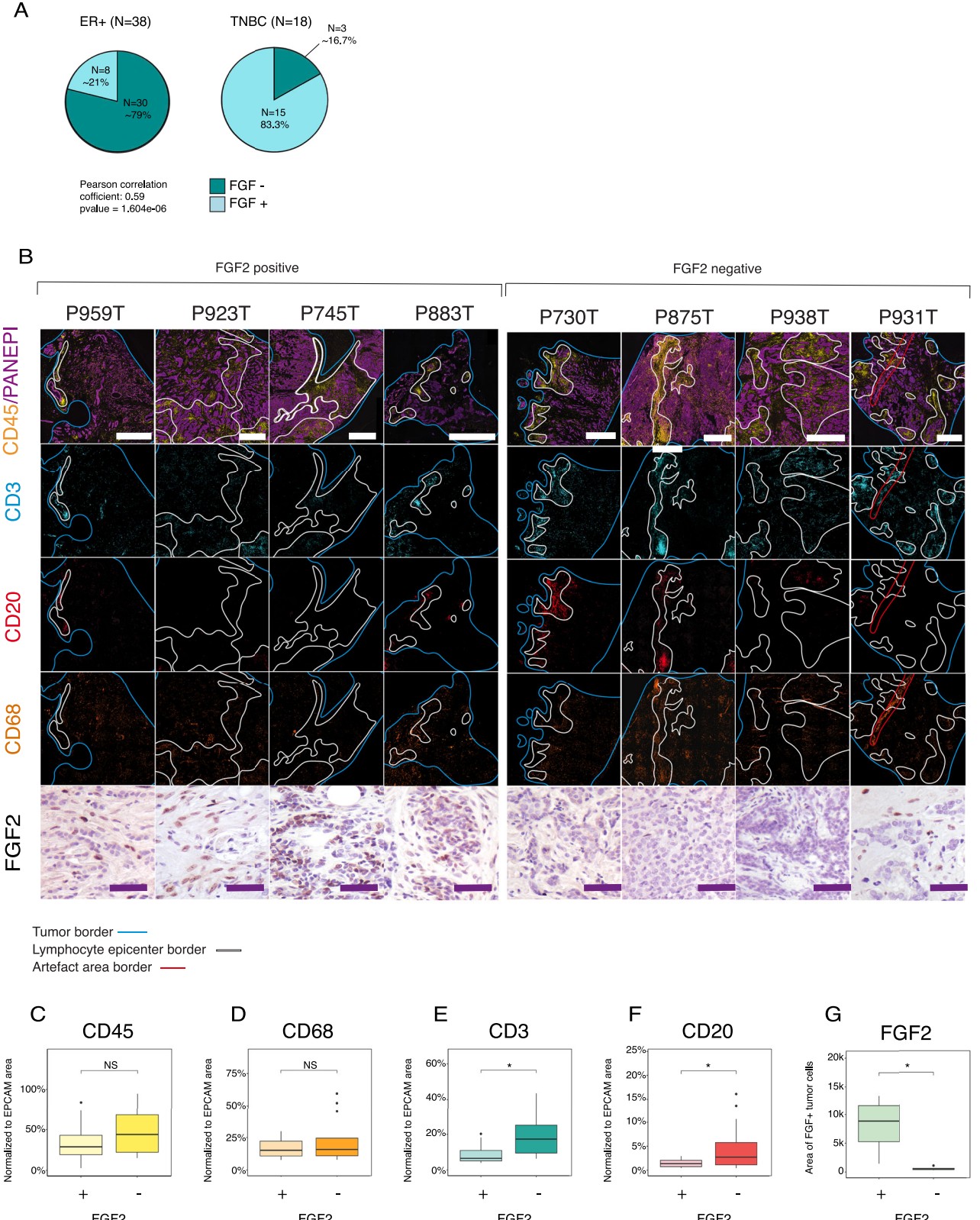

**Fig. 5 | FGF2 enriched tumors have less T-cell and B-cell infiltrates.**
**A** Immunohistochemically (IHC) stained fibroblast growth factor 2 (FGF2) expression of 45 primary breast cancer samples representing either estrogen receptor (ER + ) or triple negative breast cancer (TNBC) subtype. The statistical significance was calculated with 2-sided Pearson's correlation test. **B** Multiplex and hematoxyline & eosine (H&E) images from 8 samples stained with CD45, EPCAM, CD3, CD20, and CD68. CD45-rich areas are encircled with white lines. The red line encircles a wrinkle in the section. Full size images are presented in Supplementary Fig. 12. Scale bars in immunofluorescent (IF) and H&E images is 100 μm. **C–G** Quantifications of CD45, CD20, CD3, CD68, EPCAM, and FGF2 positive areas from IHC and IF staining. The expressions of CD45, CD20, CD3, and CD68 was normalized to the size of the tumor tissue (cells/tumor area). Three individual measurements were taken from each image (IF) and one from FGF2 IHC. The statistical significance was calculated with one-sided t-test. Exact p-values are 0.0096 for CD3, 0.033 for CD20 and 0.03 for FGF2.

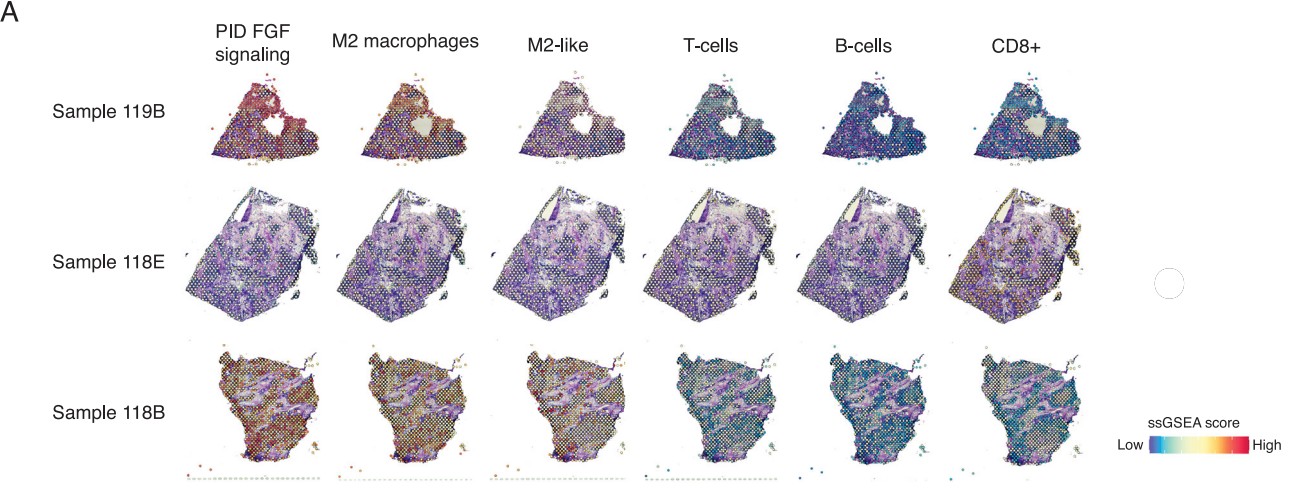

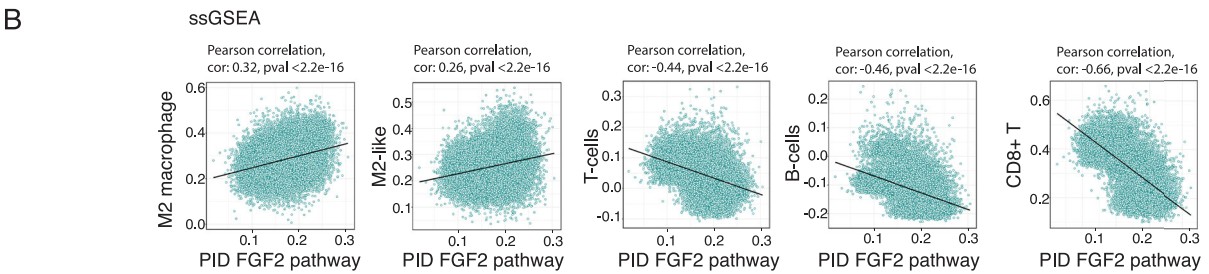

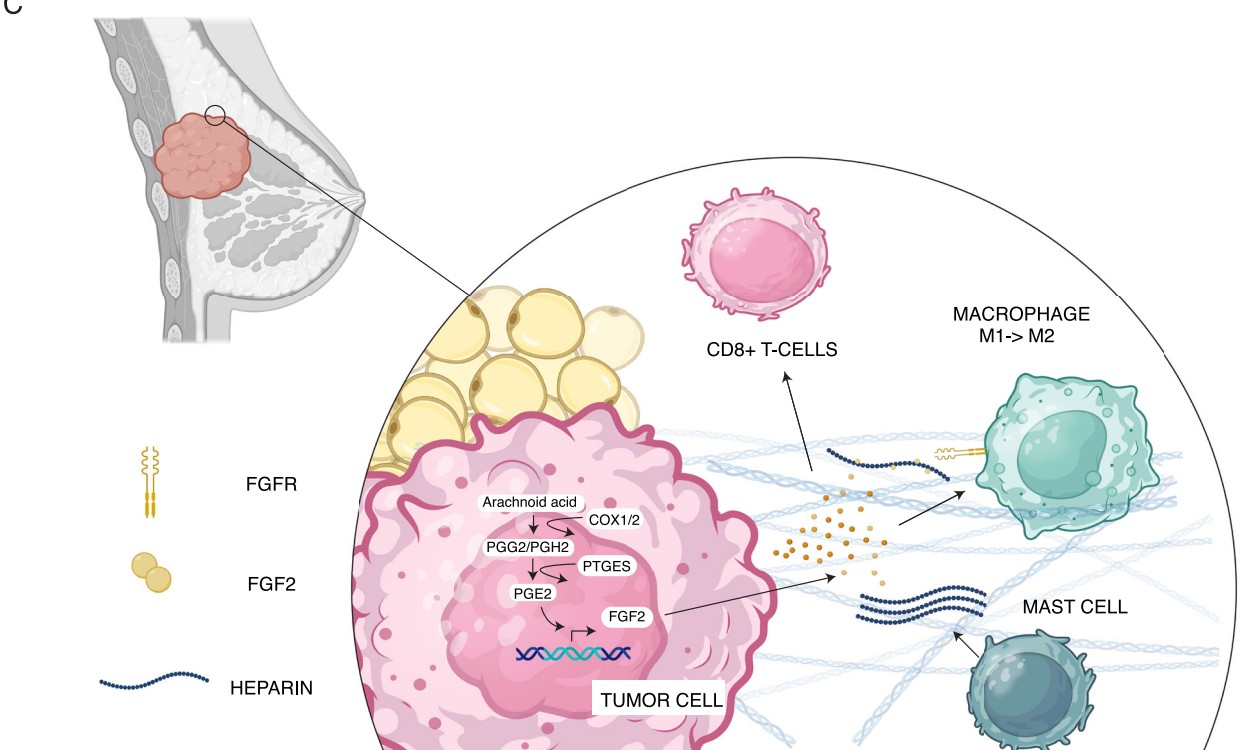

**Fig. 6 | FGF-signaling rich areas have more M2-macrophages and less CD8 + T- and B-cells. A** Spatial feature blots of 9 individual samples representing the expression of fibroblast growth factor (FGF) -signaling, as well as M2-markers and T and B-cell specific markers (ssGSEA). A full list of gene sets is provided in the Source Data file and all images are present in Supplementary Figs. 8 and 9. **B** Two-sided Pearson correlation was performed between PID_FGF_Pathway and M2, T and B-cell specific signatures defined by ssGSEA. **C** Schematic representation of the model. Created in BioRender. Peura, A. (2025). https://BioRender.com/0yf2j1z.

proliferation with ki67 (Abcam, ab15588), for apoptosis with cleaved caspase 3 (CC3) (Cell signaling # 9661), for cytoskeleton with F-actin (AlexaFluor Phalloidin 546 Invitrogen, A22283), and Hoechst 33258 (Sigma) for nuclei. On day 7, PDECs were fixed with 4% paraformaldehyde (PFA) for 10 min and washed once with PBS. Samples were permeabilized with 0.25% Triton-X in PBS for 20 min and washed with IF buffer containing 7.7 mM $NaN_2$, 0.1% BSA (Biowest),0.2% Triton-X and 0.05% Tween. Next, the samples were treated with 10% normal goat serum (cat. no 16210064, Thermo Fisher Scientific) in IF-buffer to block the nonspecific binding sites for 1 h at room temperature. The primary antibody was diluted into the blocking buffer and incubated overnight at +4 °C. The next day the unbound primary antibody was washed away with IF-buffer (3 x 20 min) with gentle rocking at RT. The secondary antibody was diluted into the blocking buffer and incubated with the samples overnight at +4 °C. Confocal laser scanning microscope (Leica TCS SP8 MP CARS with a HC PL APO 40x/1.10 water CS2 objective, Leica Microsystems, Biomedicum Imaging Unit, HiLIFE, University of Helsinki and Biocenter Finland) was used for imaging the structures. The pinhole was set to 1.00, AU and 405 nm diode, 488 nm argon and 561 nm DPSS lasers were used. List of antibodies used is available in Supplementary Information file as Supplementary table 1.

## Cytokine profiling
The growth medium was collected from PDEC samples on day 3 and stored at −80 °C until analysis. After thawing supernatants, the remains of cells and debris were removed by centrifugation (10 min, 10,000 g, RT). Cytokines secreted into the growth medium were analyzed with R&D DuoSetELISA kits (IL-10: DY217B-05, IL-6: DY206-05, IL-1β/IL1F2: DY201-05, LAP (TGF-β1): DY246-05) together with DuoSet Ancillary Reagent Kit 2 (DY008) according to manufacturer's instructions.

## Flow cytometry
PBMCs and PDEC samples were grown in 3D matrices (GrowDex 0.3%, GrowDex 1.0%, PeptiGel, and Matrigel). NC matrices were dissociated by adding 1:2 GrowDase (UPM Biomedicals) enzyme diluted in Mammocult and incubating samples at +37 °C for 1 h. PeptiGel was degraded with Cell recovery solution (Corning) at +4 °C for 1 h or with Pronase (Sigma) solution 10 mg/mL in PBS for 10 min at +37 °C. Matrigel was enzymatically digested by adding 400 μL of undiluted Cultrex Organoid Harvesting Solution (Nordic Biosite, 3700-100-01) at +4 °C for 1 h. After enzymatic dilution of the growth matrix, the cells were separated by using a filter cap tube (Falcon™) and centrifugating the cells at 400 g for 5 min at +4 °C.

Harvested cells were stained with the primary antibodies at +4 °C for 1 h in a staining buffer containing phosphate buffered saline solution (PBS), 2 mM ethylenediamine tetra acetic acid (EDTA) (Cat: E9884, Sigma-Aldrich), and 1% heat-inactivated FBS (Serana). CountBright Absolute Counting Beads (Invitrogen, C36950) were used for cell number counting. The samples were sorted with either BD FACS Aria II (BD Biosciences) or with NovoCyte Quanteon. UltraComp eBeads Plus Compensation Beads (Invitrogen, 01-3333-42) were labeled with single conjugated antibodies to set the signal compensation. The data was analyzed with FlowJo version 10.8. The gating strategies and antibody panels are shown in the Supplementary Fig. 11. List of antibodies used is available in Supplementary Information file as Supplementary table 1.

## Bulk-mRNA sequencing
PDEC samples were cultured for 7 days in 0.3% NC, 1.0% NC, and in Matrigel. The samples were collected for RNA sequencing by centrifugating the 3D cultures (300 g) without enzymatic dissection and snap freezing the pellet at −80 °C. The uncultured samples were frozen directly after overnight collagenase treatment. The total RNA was isolated by using RNeasy Mini kit (Qiagen 74106) and the extracted RNA was cleaned with Zymo RNA clean & concentrator kit (R1019)

according to the Qiagen's instructions. RNA concentration was measured with Qubit® 2.0 Fluorometer (Invitrogen) and the RNA quality was measured with TapeStation (Agilent). RNA was sequenced with Illumina NextSeq 500. The service was provided by the Biomedicum Functional Genomics Unit at the Helsinki Institute of Life Science and Biocenter Finland at the University of Helsinki.

The BRB-sequencing method was based on the Drop-seq protocol described in Macosko EZ et al.[63]. First the RNA samples (10 ng) were barcoded using Indexing Oligonucleotides (Integrated DNA Technologies). cDNA was prepared from RNA samples (10 ng) using RT mix containing Maxima RT buffer, 1 mM dNTPs, Maxima H-RTase (all ThermoFisher Scientific) and Template Switch Oligo (Integrated DNA Technologies). RiboLock (ThermoFisher Scientific) was used to inhibit the Rnases. The samples were incubated in T100 thermal cycler (BioRad).

The cDNA was amplified by PCR using RT mix as template, 1x HiFi hotStart Readymix (Kap) Biosystems) and SMART PCR primer. The samples were thermocycled in a T100 thermocycler (BioRad). The PCR products were pooled together in sets of 12 samples containing different Indexing Oligos and purified with 0.6X Agencourt AMPure XP Beads (Beckman Coulter) according to the manufacturer's instructions. The purified cDNA was tagmented using the Nextera kit. The reaction was performed according to manufacturer's (Nextera) instructions, apart from the P5 SMART primer that was used instead of S5xx Nextera primer.

The concentration of the libraries was measured using a Qubit 2 fluorometer (Invitrogen) and the Qubit DNA HS Assay Kit (ThermoFisher Scientific). The quality of the sequencing libraries was assessed using the LabChip GXII Touch HT (PerkinElmer), with the DNA High Sensitivity Assay (PerkinElmer) and the DNA 5 K / RNA / Charge Variant Assay LabChip (PerkinElmer). The libraries were sequenced with a Illumina NextSeq 500, with a custom primer producing read 1 of 20 bp and read 2 (paired end) of 50 bp. The sequencing was performed at Biomedicum Functional Genomics Unit (Fugu).

Oligonucleotide sequences:

TSO: AAGCAGTGGTATCAACGCAGAGTGAATrGrGrG SMART PCR primer: AAGCAGTGGTATCAACGCAGAGT

P5 SMART primer:

AATGATACGGCGACCACCGAGATCTACACGCCTGTCCGCGGAA GCAGTGGTATCAA

CGCAGAGT*A*C

Sequencing read 1: GCCTGTCCGCGGAAGCAGTGGTATCAACGCA GAGTAC

The data was normalized and top differentially expressed genes were defined by using DeSEQ2[64] and results were verified by using edgeR[65,66]. Top differentially expressed pathways were defined by GSEA v4.2.3[21], gsva or by FGSEA[67,68]. FDR < 0.2 or pval < 0.05 (stated separately) were considered statistically significant. Gene sets used for analysis are visible in supplementary table 1. Visualizations were made by either R v4.1.1 and R-studio or Cytoscape v3.9.1[69] with Enrichment map plugin[70].

The data for the gene expression comparisons used in Fig. 3D was obtained from publication by Munne et al. 2021 and processed as described in the publication[12].

## qRT-PCR analysis
Samples for the qRT-PCR analysis were cultured as previously stated in the PBMC-culture section. The total RNA was isolated by using RNeasy Mini kit (Qiagen 74106) and the extracted RNA was cleaned with Zymo RNA clean & concentrator kit (R1019) according to the Qiagen's instructions. The cDNA synthesis was performed by BioRad T100 thermal cylcer, with iScript Reverse Transcription Supermix (BioRad). Real-time RT-PCR was performed with Light-Cycler® 480 II (Roche) using DyNAmo ColorFlash SYBR Green (Thermo Scientific).

## Single-cell RNA sequencing

Two fresh tumor samples were dissociated with collagenase A (1 mg/mL; Sigma) containing the MammoCult media (StemCell technologies) with gentle shaking (130 rpm) at +37 °C overnight. Next the samples were divided into two; one part was cultured in 0.3% NC for 7 days and another part was processed for flow cytometry sorting. The cells were immunolabeled with CD45 antibody and sorted with BD Influx Cell sorter at the HiLIFE Flow Cytometry Unit (University of Helsinki). After sorting, the cells were collected in $Mg^{2+}/Ca^{2+}$ depleted PBS and the sequencing was done at the FIMM Single-Cell Analytics unit (University of Helsinki and Biocenter Finland, Finland). After 7 days of ex vivo culturing, the matrix was dissociated with GrowDase and the CD45 positive cells were sorted as described above.

Single cell sequencing was performed by Finnish Institute for Molecular Medicine (FIMM). Gene expression of single cells was defined by using 10x Genomics Chromium Single Cell 3'RNAseq platform and the run and library preparation were done using the Chromium Next Gem Single Cell 3' Gene Expression, v3.1 Dual Index chemistry. Prepared sample libraries were sequenced on Illumina NovaSeq 6000 system (Illumina) using read lengths 28 bp (Read 1), 10 bp (i7 Index), 10 bp (i5 Index) and 90 bp (Read 2). Data preprocessing, alignment to hg38, and aggregation was done with 10x Genomics Cell ranger v4.0 (10x).

Afterwards, samples were imported into R-studio and following analysis was performed with Seurat v3[71–73]. Individual cells were filtered with the following criteria to obtain healthy and reliable cells without any doublets: cells with more than 10% of mitochondrial genes were filtered out. Cut off for number of individual genes was set above 200 and under 5000, cutoff for unique molecular identifier (UMI) above 500 and under 30,000. After removal unhealthy cells and doublets, a total of 12,188 cells were selected for further analysis. Seurat v3[71–73] was used for integration of samples to minimize the batch effect between different samples. The filtered gene barcode-matrix was then normalized with Seurat v3 log2 based normalization.

Finally, the gene-barcode matrix was analyzed with principal component analysis (PCA) and Uniform Manifold Approximation and Projection (UMAP) was performed with top 20 principal components. Optimal number of PCA's were selected by using JackStraw-analysis and with visual representation of top genes in each cluster. The graph-based clustering at the resolution of 0.5 was performed following guidelines of Seurat v3 library. Optimal resolution was obtained with clustree-package and visual interpretation[74]. Individual clusters were identified with conventional markers for each cell type.

Pseudobulk-data was created from total unnormalized mRNA-counts by calculating the sum of counts for each sample. Normalization and differentially expressed genes between culture conditions were defined by using EdgeR[65,66] TMM normalization. Patient identity was used as a blocking term. For fast pre-ranked gsea analysis, pseudobulk-data was calculated as described earlier. For obtaining the differences of individual patients, differentially expressed genes per patient were calculated by wilcox single-rank sum test. The list of gene-sets used for FGSEA are provided in source data file.

## Immunohistochemistry (IHC)

Tissues were fixed with 4% PFA and embedded in paraffin. The samples were sectioned into 5 µm slices and deparaffinized. The heat-induced antigen retrieval was performed at +60 °C in a citrate buffer solution (Dako) overnight. IHC staining was carried out using standard techniques. Primary antibody for FGF2 (ab208687, Abcam) was used at the concentration of 1:200. Images were taken with a Leica DM LB microscope (Biomedicum Imaging Unit, University of Helsinki, Finland).

The multiplex immunohistochemistry was performed at the FIMM Digital Microscopy Unit. List of antibodies used is available in Supplementary Information file as Supplementary table 1. The quantification of CD45, CD3, EPCAM, CD20, and CD68 was done with Adobe Photoshop 2024 by selecting all cells with marker and calculating the area. Each marker was analyzed three times by selecting three different cells as a starting point. The average FGF2 expression of each sample was analyzed similarly, but only one measurement per sample was taken as some of the samples contained less than three FGF2 positive cells.

## Live imaging and light microscopy

PDECs were stained with CD45+ VioBright (Miltenyi Biotech) antibody 1:100 for 1 h at +37 °C. Next the samples were washed twice with PBS before adding the growth media. Images were taken with Nikon Ti Eclipse, 1 minute frequency for 5 hours, 20x magnification and with Plan Apo λ 20x Ph2 DM and Cy5 lamp power 60. Samples were kept in humidified incubator during the imaging (Biomedicum Imaging Unit, University of Helsinki, Finland).

## Western blotting

Protein samples were isolated using a RIPA lysis buffer supplemented with protease (Roche) and phosphatase inhibitors (Roche). The nuclei were broken with a 20 G needle. The concentration of the isolated proteins was measured with the BCA Protein Assay Reagent (BioRad). Ten to twenty micrograms of protein were separated on the BioRad gradient gels (4% to 20%) and transferred on the nitrocellulose membranes (BioRad). The membranes were then incubated with the primary and secondary antibodies according to the manufacturer's recommendations. List of antibodies used is available in Supplementary Information file as Supplementary table 1.

## Spatial transcriptomics

Samples were downloaded from Gene Expression Omnibus (GEO)[75] with accession number GSE210616. In the analysis we followed the analysis performed by the authors Bassiouni et al. in the original publication[53]. First, we normalized the data with Seurat SCTransform-normalization. For further analysis, only samples with some FGF2 expression and no prior neoadjuvant treatment were selected. Afterwards, we defined the tumor purity, stromal and immune score with ESTIMATE[76]. For further analysis, only spots containing 0.75 or more of the tumor-derived reads were retained. Then, ssGSEA2 was used with FGF-associated gene sets (msig-database), as well as macrophage associated signatures (source data file) and T and B-cells signatures from the original publication, and the association between FGF2 signaling and the cell type derived reads were calculated by Pearson correlation.

## Statistics and reproducibility

Results are reported as medium +- standard deviation unless otherwise specified. All the experiments were done at least in three technical repeats unless otherwise stated. The boxplots are always representing the following unless otherwise stated: the box shows the interquartile range (IQR), which contains the middle 50% of the data: the bottom of the box marks the first quartile (Q1), and the top marks the third quartile (Q3). Inside the box, a line indicates the median (Q2), or the middle value of the dataset. The "whiskers" extend from the box to the smallest and largest values that are within 1.5 times the IQR from the quartiles. Data points outside this range are considered outliers and are typically shown as individual dots. In the figures stars represent following p-values unless otherwise stated: *<0.05, **<0.01, *** < 0.001, **** < 0.0001. We have added the version of all key packages to the end of the source data.

## R-studio and libraries used for the study

R-studio v. RStudio 2023.06.1 + 524 "Mountain Hydrangea" and R version 4.3.1 (2023-06-16) were used for analysis and visualization.

## Reporting summary

Further information on research design is available in the Nature Portfolio Reporting Summary linked to this article.

## Data availability

The scRNA seq data has been deposited to Zenodo and can be accessed via following link: https://doi.org/10.5281/zenodo.11118899. The bulk mRNA seq data is provided in the source data. The publicly available data from Bassioni et al. is available in GEO database under accession code GSE210616. The publicly available data from Munne et al. is available in Sequence Read Archive (SRA) database and are accessible through the SRA accession numbers: PRJNA663587. Source data are provided with this paper. The remaining data are available within the Article or Source Data file. Source data are provided with this paper.

## Code availability

The code used in the Fig. 6B is available in the Zenodo and can be accessed via following link: https://doi.org/10.5281/zenodo.11118899.

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

## Acknowledgements

We are grateful to the patients who participated in this study and made it possible, and to the personnel at Helsinki University Central Hospital who assisted with the recruitment of patients and collection of the sample material. We thank the Biomedicum Functional Genomics Unit (FuGU, HiLIFE, UH and Biocenter Finland) for their genome profiling services and the Biomedicum Imaging Unit (BIU, HiLIFE, UH, and Biocenter Finland) for the microscopy support. The authors would also like to thank FIMM Single-Cell Analytics unit supported by HiLIFE and Biocenter Finland for the single-cell analytics services. We also want to thank Annabrita Schoonenberg (FIMM Digital Microscopy and Molecular Pathology Unit supported by HiLIFE and Biocenter Finland) for performing the multiplex IHC staining. This study was funded by UPM Biomedicals and by grants from the Academy of Finland (JK), the Finnish Cancer Organization (JK), the Sigrid Juselius Foundation (JK), and Finnish Cancer Institute (FCI)(JK), Finnish Medical Foundation (AP), Jane and Aatos Erkko Foundation (JK), Biomedicum Helsinki Foundation (AP) and Eemil Aaltonen Foundation (AP). M. Heilala acknowledges funding from the Academy of Finland's Flagship Program under Projects No. 318890 and 318891 (Competence Center for Materials Bioeconomy, FinnCERES) and Finnish Foundation for

Technology Promotion. Moreover, the work was funded by the Business Finland R2B funding (2489/31/2017, Preclinica; 42533/31/2020, Immunate), and the RESCUER project, which has received funding from the European Union's Horizon 2020 Research and Innovation Programme under Grant agreement No. 847912. This work was also supported by the CDMRP W81XWH211-0773/-0774. Opinions, interpretations, conclusions, and recommendations are those of the author and are not necessarily endorsed by the Department of Defense. Illustrations 1 A, 4 A and 6 C were created by BioRender. Open access was funded by Helsinki University Library.

## Author contributions

Study design: A.P., R.T., R.L., M.S., M. Heilala., P.M., J.A., P.M.M.; writing group; A.P., J.K., T.A.T., P.M.M.; data analysis: A.P., R.T., R.L., M.S., P.M.M., M. Hollmén.; bioinformatic analysis: A.P.; provided patient material and clinical data/analysis: P.M., T.M., L.N., M.M., P.E.K., J.M., P.H., P.P., T.K., M.K., O.U.; All authors read and approved the final version of the paper.

## Competing interests

The authors declare no competing interests.

## Additional information

[1]Cancer Cell Circuitry Laboratory, Translational Cancer Medicine Research Program, Research Programs Unit, & Medicum, University of Helsinki, Helsinki, Finland. [2]Medicity Research Laboratory, University of Turku, Tykistökatu 6A, Turku, Finland. [3]Department of Applied Physics, Aalto University, Espoo, Finland. [4]Finnish Genome Editing Center, HiLIFE infrastructures, University of Helsinki and Biocenter Finland, Helsinki, Finland. [5]UPM Biomedicals, UPM-Kymmene Corporation, Helsinki, Finland. [6]Department of Pathology, HUSLAB and Haartman Institute, Helsinki University Central Hospital and University of Helsinki, Helsinki, Finland. [7]Breast Surgery Unit, Comprehensive Cancer Center, Helsinki University Hospital and University of Helsinki, Helsinki, Finland. [8]Comprehensive Cancer Center, University of Helsinki & Helsinki University Hospital, Helsinki, Finland. [9]Department of Surgery, Kymenlaakso Central Hospital, KYMSOTE, Kotka, Finland. [10]Finnish Cancer Institute, Helsinki, Finland. [11]FICAN South, Helsinki University Hospital, Helsinki, Finland. [12]Department of Cell & Tissue Biology, University of California, San Francisco, 513 Parnassus Avenue, UCSF Campus, San Francisco, CA, USA. [13]These authors contributed equally: Rita Turpin, Ruixian Liu. [14]These authors jointly supervised this work: Juha Klefström, Pauliina M. Munne. ✉e-mail: juha.klefstrom@helsinki.fi; pauliina.munne@helsinki.fi

