## [Transparent Peer Review file · Nature Communications]

Soft Matrix Promotes Immunosuppression in Tumor-Resident Immune Cells via COX-FGF2 Signaling

Corresponding Author: Professor Juha Klefstrom

Version 0:

Reviewer comments:

Reviewer #1

(Remarks to the Author)

In this manuscript the authors investigated the impact of matrix stiffness on TIL composition and activity in patient-derived explant cultures (PDECs). The authors previously reported that compressive stress is an essential element of the ER phenotype in breast cancer using PDEC cultures.

In this report the authors report that soft matrix do not alter the composition of tumor infiltrating leucocytes (TILs) directly. Instead soft matrix induces up regulation of COX2 pathway activity and FGF2 secretion. The authors cultivate PDEC explants in different nanocellulose (NC) concentration matrices in order to investigate the impact of the matrix stiffness onto the composition of the microenvironment.

One of the bias introduced by this read out is the generation of matrix stiffness onto an already organized microenvironment. Modification of the matrix stiffness is the result of extracellular matrix deposition by cancer associated fibroblasts and it is amplified by intratumoral tumor associated macrophages. Therefore here only the impact of compressive forces from tissue are evaluated and not the stiffness generated by recruited/and or resident cells. The title of the manuscript do not reflect the data reported.

Another important point is the use of PBMC "uncultured" as a control for the PDEC explant cultured in soft condition. PBMC contain a variety of cells that do not recapitulate the tumor infiltrating immune cells. For instance, CD8+ T cell (a key element of the anti-tumoral response) are only recruited to the tumor microenvironment if they are antigen specific and activated, the frequency of those antigen specific CD8+ T cell in the PBMC being very low introducing a bias to the interpretation of CD8+ T cell diminished population in the soft conditions. Likewise the increased of M2-like macrophages signature in soft condition compared to "uncultured" PBMC, as a high proportion of macrophages are differentiating in situ after entering the tumor microenvironment.

Several specific points need to be addressed:

1. In figure 2 the authors shown that the CD8+ T cells are lost in soft-matrix condition compared to "uncultured PBMC" condition and conclude that soft matrix conditions lead to that loss. What about stiff matrix condition? The uncultured PBMC is one of the controls but not the one that permits to state on the impact of soft matrix conditions. The reference should be the stiff matrix conditions as illustrated in Fig.1. The authors argue that is challenging to remove stiff matrix for flow cytometry. Nevertheless, they show data in Fig.1C that prove the feasibility to perform FACS analysis. This comparison to stiff as defined to soft is important in order to correctly interpret the data. In Fig. 3G data on mRNA sequencing are presented in soft, stiff and uncultured PBMC. The stiff conditions should be included in the analysis of Fig.2 in order to conclude on the impact of soft condition on the immune cell reprogramming. In order to conclude that the matrix stiffness is affecting the proportions of CD8+ T cells, the authors should compared the soft matrix condition to stiff matrix conditions.

The up-regulation of M2 genes is not documented. There is no supplementary Table 1. This table contains the list of genes up/down regulated M1/M2 genes.

From the data reported in Fig. 2 G, H and I there is downregulation of the M1 like phenotype but there is no clear data on up-regulation of a M2-like phenotype (data missing and no commented).

3. In Fig3. the authors report that FGF2 and PTGS2 genes are regulated in soft conditions once again compared to uncultured samples. The appropriate control should be NC stiff conditions.

4. In Fig.4 the authors use a model of in vitro differentiated macrophages from PBMC to test the induction of an M2-like phenotype. They report increased CD163 expression and decreased CD86. All the macrophages express CD163 (% of positive cell vs MFI on whole population)? Several other markers are important to defined an M2- like phenotype for instance expression of MHC cl II molecules, CD80 but also Arg1, iNOS. What is the expression of those markers in this context? This is important in order to define an "M2" like phenotype of in vitro generated macrophages.

5. The manuscript have several typos et mistakes in the labelling of the supplementary legends : Supplementary Fig. 3, Supplementary Fig. 6

Reviewer #2

(Remarks to the Author)

The manuscript by Puera et al. aims to examine the role of tissue mechanics on the phenotypes of tumor-infiltrated leukocytes (TILs). The authors use primary patient-derived explant culture (PDEC) from patients with breast cancer which is a unique resource for the study. Likewise innovative assays, such as scRNAseq, spatial transcriptomics, and multiplex IHC have been used. The topic is timely and of interest to cancer research community; however, the conclusions of the paper are very diffuse and the individual findings are not well connected to one another. No direct functional effect of stiff versus soft microenvironments on TIL function has been demonstrated. Most importantly, the experiments do not directly compare PDECs in soft versus stiff matrices. Instead, uncultured PDECs are compared to PDECs in soft matrices making the results difficult to interpret.

The basis for the study is the notion that tumor-infiltrating leukocytes move from a stiff to a soft microenvironment. Although this is an interesting hypothesis, there is no evidence that this actually occurs. I appreciate that this would be difficult to measure the stiffness changes in vivo, but stating that cells that border a necrotic region are exposed to a sudden relief of ECM-generated pressure is an overstated claim. I think it is safe to say that cancer cells and immune cells are exposed to gradients of stiffness over the course of tumor progression.

However to support the conclusion that "stiffness regulates antitumor immunity" stiff versus soft culture conditions should be compared. Nearly all of the assays compare uncultured PDECs to PDECs in a soft matrix. It is unclear whether uncultured PDECs are an appropriate substitute for PDECs in a stiff matrix. Is there evidence to support that PDECs cultured in a stiff matrix are equivalent to PDECs that have never been cultured? I appreciate that the authors state that isolating cells from the stiff matrices was unsuccessful/difficult, but alternate assays such as sectioning and staining could be pursued. At the very least in assays (such as the cytokine assay) where PDECs in stiff cultures can be assessed, uncultured PDECs should be included as a comparison to stiff cultures.

It is unclear whether the changes in RNA expression and release of cytokines are occurring in the cancer cells or the TILs (see Figure 1i-k). The authors conclude that since PBMCs cytokine levels are undetectable that "matrix alone without tumor tissue is not sufficient to induce production of these cytokines from immune cells". An alternative hypothesis would be that cancer cells (not immune cells) are producing cytokines. Can the scRNAseq (Figure 2) data address this question? For example, can the cancer cells and immune cells can be interrogated separately?

It is unclear why the FGF and COX2 pathways were considered for follow-up to the exclusion of all other pathways that were enriched. Is FGF2-FGFR or COX2 known to cause the release of immunosuppressive cytokines identified in Figure 1?

Figures 3-4 are divergent from the main focus of the paper. Soluble FGF2 is added to CD14+ cells isolated from peripheral blood and this forces them undergo M2 macrophage polarization. This has been demonstrated previously in transgenic mouse models with altered FGF2 expression? But how does this connect back to stiff/soft matrices? Do PDECs in soft matrices produce more FGF2 protein (not mRNA) than PDECs in stiff matrices? Can the secreted media from PDECs in soft matrices (vs stiff matrices) cause M2 macrophage polarization?

I disagree with the statement that stromal cells are nearly negative for FGF2 staining. See P923T, P959T, P840T, P747T. What were the controls were used to develop the IHC protocol for FGF2? How does FGF2 staining in tumor samples connect back to stiffness-induced changes? Was the FGF2 staining stronger in regions of lower compression?

Minor comments:

Figure 1c-d. Please comment on what the color gradient is intended to quantify.

Figure 1g. The scaling is different for each image making it hard to compare.

Figure 2. Legend typo ... "patients were cultured in soft matrices" should be PDECs were cultured in ...

Fig 4C is labeled in the figure as CD86 but referenced in the text to be a measurement of Heparin

Fig 3G . The text states the figure compares mRNA expression of several genes in PDECs that were uncultured, cultured in stiff matrices, or in soft matrices. The figure does not display results for uncultured cells.

I can not provide a critique for RNA sequencing pipelines and analysis.

Version 1:

Reviewer comments:

Reviewer #1

(Remarks to the Author)

The authors respond to all questions raised.
There are still 2 pending questions that were not properly answered here below.

Pending questions

1. Authors' response:

The single cell sequencing was performed on two patient samples, P981T and P982T. The results showed depletion of M1 signatures in both patient samples. The P981T demonstrated also upregulation of M2 signatures whereas the P982T sample showed decrease in relative M1 to M2 signature, but we could not find significant M2 signature in P982T in the scRNASeq data. This could be due to resolution issues related to lower immune cell count in P982T sample (see Figure 2B). To confirm the observed upregulation of M2 macrophages in soft matrix cultures, we analyzed macrophage phenotypes from three additional patient samples with flow cytometry. The patient samples were grown as PDECs in soft and stiff (PG) for three days before analyzing the macrophage markers. The new Figure 2J shows the upregulation of CD163 and CD206 marker of M2 macrophage in all the patients. The new data have been added to the revised manuscript as Figure 2 J and new text has been added accordingly (rows 226-233).

Reviewer 1 response

The authors have now 3 new patients included. The data from the 3 patients do not show statistically significant differences in terms of expression on CD206 and CD163.
Why there is no analysis of CD8+ T cells in the all three patients as provided for macrophages in order to appreciate if there is any statistically differences?

2. Author's response

To quantify the phenotype of PBMC-derived macrophages after FGF2 + heparin treatment, we assembled a qRT-PCR primer panel, containing a set of M1 and M2 markers, including Arg1, iNos2, and CD80 pointed out by the reviewer. The panel was based on publications (PMID: 29961579, PMID: 35325594, PMID: 37322116). The mRNA expression of CD80 was not significantly altered between the control and FGF2 + heparin treated macrophages. As we did not detect Arg1 or iNos with any of the primers used, we also performed the western blot analysis for these markers and concluded that PBMC-derived macrophages express low or undetectable levels of Arg1 and iNOS. The negative western blot results are shown below and intended for the reviewer only. Thus, we are unable to conclusively define whether these markers are significantly altered between the control and FGF2 + heparin treated macrophages. However, from our qRT-PCR panel we were able to conclude that M2-related macrophage marker (CCL17) was upregulated with FGF2 + heparin treatment, while M1-related macrophage marker (CXCL9) was downregulated. We hope that the CCL17 and CXCL9 macrophage polarization markers, which were consistently changed in three PBMC samples, provide sufficient evidence to support M2-polarization of FGF2 + Heparin treated macrophages.

The reason why we did not observe changes or expression of some of the markers pointed out by the reviewer is most likely due to differences in the macrophage origin. In vitro differentiated macrophages from PBMCs are slightly different from tissue derived macrophages as they originate from blood monocytes and not from the primitive erythro-myeloid progenitors (EMPs) as the tissue derived macrophages (PMID: 24854589, PMID: 31189095). Therefore, some of the most common M2-markers and TAM-markers may be differentially expressed between the PBMC-derived and tissue-derived macrophages.

Reviewer 1 response

The authors claim that In vitro differentiated macrophages from PBMCs do not express M2 markers. They do not express it if there are not polarized into an M2 phenotype. Therefore a control (using anti-IL4 and IL-13) should be used to compare the

results with FGF2+Heparin treated macrophages.

Reviewer #2

(Remarks to the Author)

The authors have done their best to address my queries/comments/concerns.
I am very satisfied with their effort and improvement of the manuscript.

Version 2:

Reviewer comments:

Reviewer #1

(Remarks to the Author)

the authors respond to all the questions.

Reviewers' comments:

Reviewer #1 (Remarks to the Author):

In this manuscript the authors investigated the impact of matrix stiffness on TIL composition and activity in patient-derived explant cultures (PDECs). The authors previously reported that compressive stress is an essential element of the ER α phenotype in breast cancer using PDEC cultures.

In this report the authors report that soft matrix does not alter the composition of tumor infiltrating leucocytes (TILs) directly. Instead, soft matrix induces upregulation of COX2 pathway activity and FGF2 secretion. The authors cultivate PDEC explants in different nanocellulose (NC) concentration matrices to investigate the impact of the matrix stiffness onto the composition of the microenvironment.

One of the bias introduced by this read out is the generation of matrix stiffness onto an already organized microenvironment. Modification of the matrix stiffness is the result of extracellular matrix deposition by cancer associated fibroblasts and it is amplified by intratumoral tumor associated macrophages. Therefore, here only the impact of compressive forces from tissue are evaluated and not the stiffness generated by recruited/and or resident cells. The title of the manuscript do not reflect the data reported.

Authors' response:

We agree that *ex vivo* culture systems cannot authentically recapitulate all factors causing extracellular matrix stiffness in cancer tissue. The reviewer is correct in pointing out that we are focusing on the impact of compressive forces, not the stiffness generated by recruited/and or resident cells. However, we wish to stress that rather than investigating the effects of multiple factors (compressive forces, extracellular matrix deposition) contributing to the microenvironmental stiffness of an already organized microenvironment *in vivo*, our paper aims to address the question of the consequences of decreased matrix stiffness/compressive forces on resident immune cell composition and response in *ex vivo* model of tumor microenvironment. As we discuss in the paper, we believe that tumor cells are likely exposed to soft microenvironments during certain critical events of tumor progression, such as invasion, necrosis, and buildup of treatment resistance. The effects of soft tumor microenvironment on tumor

infiltrated leukocytes and local immunity are currently unclear and therefore, we found this interesting area to research, initially using reductionist *ex vivo* models of the soft tumor immune microenvironment. To address the justified concern of using too broad title, we changed the title from “Stiffness Regulates Breast Cancer Antitumor Immunity via COX2- FGF2 Pathway” to “Soft Matrix Promotes Immunosuppression in Tumor-Resident Immune Cells via COX2-FGF2 Signaling”. We also adjusted the introduction of the manuscript on rows 68-72, “While the chemical signaling between...”.

Another important point is the use of PBMC "uncultured" as a control for the PDEC explant cultured in soft condition. PBMC contain a variety of cells that do not recapitulate the tumor infiltrating immune cells. For instance, CD8+ T cell (a key element of the anti-tumoral response) are only recruited to the tumor microenvironment if they are antigen specific and activated, the frequency of those antigen specific CD8+ T cell in the PBMC being very low introducing a bias to the interpretation of CD8+ T cell diminished population in the soft conditions. Likewise the increased of M2-like macrophages signature in soft condition compared to "uncultured" PBMC, as a high proportion of macrophages are differentiating *in situ* after entering the tumor microenvironment.

Authors' response:

In this study, we used uncultured original tumor samples and *ex vivo* stiff matrix grown PDECs (new revised version) as a comparison to soft matrix grown PDECs to investigate the impact of matrix stiffness on tumor resident immune cells. We used PBMCs only as controls in Figure 1J and Supplementary Fig 3C. These experiments demonstrate that the matrix itself does not induce *IL-10*, *TGF- β* , *IL-6*, and *IL-1 β* cytokine response in the isolated leukocytes. Since these cytokines were altered in PDEC cultures, these experiments led us to further investigate the role of the epithelial-immune cell interface in immunosuppression. Thus, we did not generally use PBMCs as a control for PDECs. We suspect that Figure 1A may have caused some confusion and therefore we have revised the figure by omitting the lower picture of PBMCs (see below). We have also adjusted the manuscript accordingly removing the references to PBMC experiments on rows 116-126 and moving them to rows 173-181, to separate more clearly the experiments using PBMCs and PDECs. We have also added the missing stiffness value of the Matrigel to manuscript row 125-126.

The old schematic (Fig 1A) and the new schematic.

In figure 2 the authors shown that the CD8+ T cells are lost in soft-matrix condition compared to "uncultured PBMC" condition and conclude that soft matrix conditions lead to that loss. What about stiff matrix condition? The uncultured PBMC is one of the controls but not the one that permits to state on the impact of soft matrix conditions. The reference should be the stiff matrix conditions as illustrated in Fig.1. The authors argue that is challenging to remove stiff matrix for flow cytometry. Nevertheless, they show data in Fig.1C that prove the feasibility to perform FACS analysis. This comparison to stiff as defined to soft is important in order to correctly interpret the data.' In Fig. 3G data on mRNA sequencing are presented in soft, stiff and uncultured PBMC. The stiff conditions should be included in the analysis of Fig.2 in order to conclude on the impact of soft condition on the immune cell reprogramming. In order to conclude that the matrix stiffness is affecting the proportions of CD8+ T cells, the authors should compared the soft matrix condition to stiff matrix conditions.

Authors' response

In the original version of the paper, we found it nearly impossible to isolate the tumor infiltrated leukocytes from the stiff matrix (NC) cultured PDECs. Supplementary Figure 3C shows PBMC isolated from the stiff matrix (NC), which was feasible because of a much higher number of leukocytes in comparison to PDECs. However, we completely agree with the reviewer that the comparison between soft and stiff matrix grown PDECs would give more reliable picture of the impact of compressive forces of matrix on the TILs and therefore, we developed a completely new matrix platform to achieve this comparison: The problem with nanocellulose (NC) was the insufficient solubilization of the matrix with cellulase enzyme (GrowDase), which hampered the

TIL isolation. We sought to replace NC with more digestible matrix and found another bioinert gel (PeptiGel - Manchester BIOGEL/CELL Guidance Systems) suitable for these purposes. PeptiGel has the same biological properties as GrowDex, but allows easier dissociation of the gel, thereby enabling the release of the cultured cells from the matrix. The gel stiffness was adjusted to match the Pa-kPa values of the GrowDex concentrations used. New data showing side-by-side comparison between soft and stiff PeptiGel (PG) matrices are presented in the new Figures (2F-G, J, 3E, Supplementary Fig 3D-F). Importantly, the new data clearly demonstrates the loss of CD8⁺ T-cells (Figure 2F-G) and enrichment of M2-like macrophages (Figure 2J). We have also removed text from the manuscript (rows 128-132) regarding the usage of the uncultured sample for a control for the stiff NC matrix as we now have a proper stiff matrix control. Moreover, we have added the data and text regarding the PeptiGel results to the manuscript (rows 196-210, 226-233, 258-261 and 272-275).

The up-regulation of M2 genes is not documented. There is no supplementary Table 1. This table contains the list of genes up/down regulated M1/M2 genes.

Authors' response:

We apologize for the missing table. The M2-gene table related to Figure 2 has been added to supplementary table.

From the data reported in Fig. 2 G, H and I there is downregulation of the M1 like phenotype but there is no clear data on up-regulation of a M2-like phenotype (data missing and no commented).

Authors' response:

The single cell sequencing was performed on two patient samples, P981T and P982T. The results showed depletion of M1 signatures in both patient samples. The P981T demonstrated also upregulation of M2 signatures whereas the P982T sample showed decrease in relative M1 to M2 signature, but we could not find significant M2 signature in P982T in the scRNASeq data. This could be due to resolution issues related to lower immune cell count in P982T sample (see Figure 2B). To confirm the observed upregulation of M2 macrophages in soft matrix cultures, we analyzed macrophage phenotypes from three additional patient samples with flow cytometry. The patient samples were grown as PDECs in soft and stiff (PG) for three days before analyzing the macrophage markers. The new Figure 2J shows the upregulation of CD163 and CD206 marker of

M2 macrophage in all the patients. The new data have been added to the revised manuscript as Figure 2 J and new text has been added accordingly (rows 226-233).

In Fig3. the authors report that FGF2 and PTGS2 genes are regulated in soft conditions once again compared to uncultured samples. The appropriate control should be NC stiff conditions.

Authors' response:

In the revision, we have performed experiments using side-by-side soft and stiff PG matrix cultures to assess changes in FGF2 and PTGS2 levels. The new data are presented in Figure 3E. We show that both FGF2 and COX2 levels are upregulated in soft matrix cultured PDEC samples in comparison to cultures grown in stiff matrix. The new Figure 3E is also shown below. The text has been revised (rows 258-261 and 272-275).

In Fig.4 the authors use a model of in vitro differentiated macrophages from PBMC to test the induction of an M2-like phenotype. They report increased CD163 expression and decreased CD86. All the macrophages express CD163 (% of positive cell vs MFI on whole population)? Several other markers are important to define an M2-like phenotype for instance expression of MHC class II molecules, CD80 but also Arg1, iNOS. What is the expression of those markers in this context? This is important in order to define an "M2" like phenotype of in vitro generated macrophages.

Authors' response:

To quantify the phenotype of PBMC-derived macrophages after FGF2 + heparin treatment, we assembled a qRT-PCR primer panel, containing a set of M1 and M2 markers, including Arg1, iNos2, and CD80 pointed out by the reviewer. The panel was based on publications (PMID: 29961579, PMID: 35325594, PMID: 37322116). The mRNA expression of CD80 was not significantly altered between the control and FGF2 + heparin treated macrophages. As we did not detect Arg1 or iNos with any of the primers used, we also performed the western blot analysis for these markers and concluded that PBMC-derived macrophages express low or undetectable levels of Arg1 and iNOS. The negative western blot results are shown below and intended for the reviewer only. Thus, we are unable to conclusively define whether these markers are significantly altered between the control and FGF2 + heparin treated macrophages. However, from our qRT-PCR panel we were able to conclude that M2-related macrophage marker (CCL17) was upregulated with FGF2 + heparin treatment, while M1-related macrophage marker (CXCL9) was downregulated. We hope that the CCL17 and CXCL9 macrophage polarization markers, which were consistently changed in three PBMC samples, provide sufficient evidence to support M2-polarization of FGF2 + Heparin treated macrophages.

The reason why we did not observe changes or expression of some of the markers pointed out by the reviewer is most likely due to differences in the macrophage origin. *In vitro* differentiated macrophages from PBMCs are slightly different from tissue derived macrophages as they originate from blood monocytes and not from the primitive erythro-myeloid progenitors (EMPs) as the tissue derived macrophages (PMID: 24854589, PMID: 31189095). Therefore, some of the most common M2-markers and TAM-markers may be differentially expressed between the PBMC-derived and tissue-derived macrophages.

We have included our new results from the qRT-PCR analysis to the manuscript Fig 4C and supplementary Fig 6E and to rows 305-310.

Western blot analysis of PBMC-derived macrophages of four donors (P1-P4), cultured with FGF2, heparin or with combination. Arg1 expression was undetectable in PBMCs. iNOS expression was very low and thus no quantification can be made, as the image below was obtained with 12 minutes imaging time and 72 hours incubation of the primary antibody.

The manuscript have several typos et mistakes in the labelling of the supplementary legends :
 Supplementary Fig. 3, Supplementary Fig. 6

Authors' response:

We thank the reviewer for pointing out the typos and mistakes in the labeling of the supplementary legends. These have been corrected.

Reviewer #2 (Remarks to the Author):

The manuscript by Puera et al. aims to examine the role of tissue mechanics on the phenotypes of tumor-infiltrated leukocytes (TILs). The authors use primary patient-derived explant culture (PDEC) from patients with breast cancer which is a unique resource for the study. Likewise innovative assays, such as scRNAseq, spatial transcriptomics, and multiplex IHC have been used. The topic is timely and of interest to cancer research community; however, the conclusions of the paper are very diffuse and the individual findings are not well connected to one another.

Most importantly, the experiments do not directly compare PDECs in soft versus stiff matrices. Instead, uncultured PDECs are compared to PDECs in soft matrices making the results difficult to interpret.

Authors' response:

We thank the reviewer for positive remarks on high interest of our work to the cancer research community and use of innovative methodology to address the question related to the impact of compressive forces of matrix to tumor infiltrated leukocytes in PDEC model. We apologize if we did not communicate the main conclusions and individual findings in a coherent way. We have tried to improve these matters in the revised version as you'll see in our point-by-point rebuttal. Most importantly, in the revised version, we have established the stiff vs soft matrix comparison by using a (*PeptiGel - Manchester BIOGEL/CELL guidance systems*), another bioinert gel suitable for explant culturing purposes. While the problem with stiff nanocellulose (NC) was the insufficient solubilization of the matrix with cellulase enzyme (GrowDase), the solubilization and cell isolation from PeptiGel allows reliable comparison between soft and stiff matrix. In this work the stiffness of the gel was adjusted to the same kPa values as the used nanocellulose. New data showing side-by-side comparison between soft and stiff PeptiGel (PG) matrices are presented in the new Figures 2F-G, 2J and 3E, as well as in Supplementary Fig 3D-F. Importantly, the new data clearly demonstrate the loss of CD8+ T-cells (Figure 2F, G) and enrichment of M2-like macrophages (Figure 2J), as well as upregulation of FGF2 and COX2 in soft matrix (Fig 3E). We have added the data regarding the PeptiGel part to manuscript (rows 196-210, 226-233, 258-261 and 272-275).

The basis for the study is the notion that tumor-infiltrating leukocytes move from a stiff to a soft microenvironment. Although this is an interesting hypothesis, there is no evidence that this actually occurs.

Authors' response:

We would like to clarify that our manuscript does not claim that lymphocytes move from a stiff to a soft microenvironment in the PDEC model. Addressing this type of question *ex vivo* would need complicated kinetic assays which have not been used in our paper. The central finding that we are reporting is the immune suppressive impact of low compressive forces of matrix on tumor cells and tumor immune microenvironment *ex vivo*, mediated via FGF2-COX2 pathway. We thought that we had communicated this clearly in the abstract: "*Our results suggest that low compressive forces in the tumor microenvironment induce local immunosuppression via phenotypic plasticity dependent FGF2 secretion from tumor cells.*" However, if there are any statements that have escaped our attention, leading to confusion, we are obviously happy to correct them. We believe

that our main hypothesis, in context of *ex vivo*, is adequately supported by the data provided in the revised manuscript.

I appreciate that this would be difficult to measure the stiffness changes *in vivo*, but stating that cells that border a necrotic region are exposed to a sudden relief of ECM-generated pressure is an overstated claim. I think it is safe to say that cancer cells and immune cells are exposed to gradients of stiffness over the course of tumor progression.

Authors' response:

We agree with the reviewer that experiments performed *ex vivo* with matrices with different compressive forces do not mirror events like sudden release in compressive stress that may happen in tumor areas undergoing necrosis. To address the reviewer's concern about the overstatement, we have more carefully worded our thoughts of the significance of *ex vivo* findings to real tumor situation: old statement: "Regardless of the cause, the cells at the borders of the necrotic regions in the tumor are exposed to a sudden relief from ECM-generated pressure.", new statement: "Tumor cells and tumor resident leukocytes are exposed to sudden or gradual softening of the tumor microenvironment, for example during necrosis, or even different gradients of stiffness over the course of tumor progression." on the manuscript row 84-86.

However to support the conclusion that "stiffness regulates antitumor immunity" stiff versus soft culture conditions should be compared. Nearly all of the assays compare uncultured PDECs to PDECs in a soft matrix. It is unclear whether uncultured PDECs are an appropriate substitute for PDECs in a stiff matrix. Is there evidence to support that PDECs cultured in a stiff matrix are equivalent to PDECs that have never been cultured? I appreciate that the authors state that isolating cells from the stiff matrices was unsuccessful/difficult, but alternate assays such as sectioning and staining could be pursued. At the very least in assays (such as the cytokine assay) where PDECs in stiff cultures can be assessed, uncultured PDECs should be included as a comparison to stiff cultures.

Authors response:

In the original version, as we explained, we encountered difficulties in isolating TILs from the stiff NC matrix due to insufficient solubilization of the nanocellulose (NC) matrix with the cellulase enzyme (GrowDase). However, in the revised version we overcame this problem by adopting the PeptiGel matrix. We have described the new experiments and data in detail in

response to reviewer's first comment (new Figures 2F-G, 2J and 3E, as well as in Supplementary Fig 3D-F, manuscript rows 196-210, 226-233, 258-261 and 272-275). The new data with side-by-side comparison between the soft and stiff PG matrix supports our main claim that low compressive forces of matrix have immunosuppressive effect on tumor immune microenvironment *ex vivo*, mediated via FGF2-COX2 pathway.

It is unclear whether the changes in RNA expression and release of cytokines are occurring in the cancer cells or the TILs (see Figure 1i-k). The authors conclude that since PBMCs cytokine levels are undetectable that "matrix alone without tumor tissue is not sufficient to induce production of these cytokines from immune cells". An alternative hypothesis would be that cancer cells (not immune cells) are producing cytokines. Can the scRNAseq (Figure 2) data address this question? For example, can the cancer cells and immune cells can be interrogated separately?

Authors' response:

We thank the referee for the relevant question. Of the cytokines investigated, TGF- β , IL-10 and IL-1 β can be produced by epithelial tumor tissue, as well as a variety of different leukocyte subtypes. However, it is often considered that the main source of IL-10 and IL-1 β are the immune cells; monocytes, macrophages, mast cell, dendritic cells, B-, and T-cells. Therefore, we inferred that the TILs in our system were the main source for at least interleukins in the soft matrix. This notion is further strengthened by our bulk sequencing analysis of the pathways upregulated in soft matrix grown PDECs, exposing mainly epithelial tissue typical pathways, but not immune function associated pathways (Fig 3A). We concluded from this that bulk-sequencing mainly identifies transcriptomic changes in epithelial cells, but the sensitivity may not be enough to determine immune specific changes. In this analysis we could not detect any other interleukin related signatures except for IL-4 and IL-12, which are known to be expressed in both epithelial and tumor cells (Fig 1H). Thus, this may indirectly imply that TILs were the main source of IL-10 and IL-1 β in the PDEC-cultures.

However, inspired by the reviewer's question, we tried to interrogate cancer and immune cells separately from the single cell data using two methods: the cytokine expression per cluster was analyzed with sum-based pseudobulked data and each cell's individual cytokine expression was analyzed with ssGSEA.

Cytokines measured per cluster with ssGSEA. The figure shows no clear difference in the cytokine expression between the original uncultured sample and the soft NC matrix.

As a third approach to address the question whether the observed immunosuppressive cytokines were produced from the tumor cell instead of the immune cells in the soft NC matrix cultures, we filtered immune cells based on their CD45 positivity from the primary tumor tissue and cultured the extracted immune cell-free tumor cells (CD45 negative) in the soft and stiff matrices as PDECs. The cytokine levels were measured with ELISA. However, unfortunately the cytokine yield was so small that we could not detect any measurable levels of IL-6, IL-10, IL-1 β , and TGF- β from any of the culture conditions. It appears that the flow cytometry sorting step in these assays yielded too low numbers of cells to for reliable cytokine measurements. The schematic of the unsuccessful experiment is shown below.

Fig: Schematic representation of the experimental set up to address whether cytokines are derived from the tumor cells or from the immune cells.

[REDACTED]

Based on these experiences, we feel that it is not trivial to explicitly define the cell type responsible for the cytokine production in PDECs. We hope that the other merits of the revised manuscript will compensate for the lack of requested data. We have added the following text to the manuscript: “As PDECs contain both tumor cells and tumor resident immune cells, it remains unclear whether the cytokines were produced by the tumor cells or by the tumor resident immune cells or both” (rows 169-171).

It is unclear why the FGF and COX2 pathways were considered for follow-up to the exclusion of all other pathways that were enriched. Is FGF2-FGFR or COX2 known to cause the release of immunosuppressive cytokines identified in Figure 1?

Authors' response:

When investigating the pathways altered in the soft matrix PDEC cultures, we observed two pathways strongly upregulated at the level of multiple gene signatures and with previously reported relation to immunosuppression, TGF- β and FGF2 (Figure 3A, B). The FGF2 signaling pathway was of particular interest, since previous report had shown that FGF2 affects macrophage programming in the mouse tumor microenvironment (PMID: 32792542). However, this pathway is overall much less studied than TGF- β pathway in immunosuppression. We had earlier observed evidence for macrophage polarization and CD8⁺ T-cell depletion and both FGF2 and COX2 were upregulated in the soft matrix cultures, which further sparked our interest to investigate this pathway. We are not aware of earlier reports linking FGF2-FGFR-COX2 pathway to immunosuppression or immunosuppressive cytokines in *ex vivo* models of breast cancer tumor immune microenvironment.

Figures 3-4 are divergent from the main focus of the paper. Soluble FGF2 is added to CD14⁺ cells isolated from peripheral blood and this forces them undergo M2 macrophage polarization. This has been demonstrated previously in transgenic mouse models with altered FGF2 expression? But how does this connect back to stiff/soft matrices? Do PDECs in soft matrices produce more FGF2 protein (not mRNA) than PDECs in stiff matrices? Can the secreted media from PDECs in soft matrices (vs stiff matrices) cause M2 macrophage polarization?

Authors' response:

We established a procedure for polarization of macrophages to study the role of COX2-FGF2 pathway in the polarization process. In this reductionist system, we demonstrate that addition of FGF2 to M0 macrophages induces polarization to M2 (Fig 4B and new Figure 4C), suggesting that this FGF2 effect on macrophages is direct, not requiring the presence of other cellular mediators. We show in the manuscript that PDECs in the soft matrix produce more FGF2 protein than in the stiff matrix (Fig 3E).

Regarding the comment on usage of a conditioned media from PDECs; the conditioned media contains many macrophage polarizing and differentiating factors and therefore, it would be difficult to pinpoint any specific factor or factors responsible for macrophage polarization. Therefore, while it is certain that there are number of factors secreted by PDECs with impact on macrophages, we rather focused solely on FGF2 in this report.

I disagree with the statement that stromal cells are nearly negative for FGF2 staining. See P923T, P959T, P840T, P747T. What were the controls were used to develop the IHC protocol for FGF2? How does FGF2 staining in tumor samples connect back to stiffness-induced changes? Was the FGF2 staining stronger in regions of lower compression?

Authors' response:

We apologize for our inaccurate statement that stromal tissue was mainly negative for FGF2 expression. Indeed, the stromal tissue was in some but not all samples strongly positive for FGF2. For example, patients P923T, P959T, P840T, and P747T, shown below. Please notice colocalization of FGF2 with mesenchymal markers Zeb1 and Vimentin, which demonstrates the mesenchymal specificity of the FGF2 staining on these samples. To accommodate the referee's comment, we have corrected the misleading sentence on the manuscript (rows 332-335).

We developed our IHC protocol by validating expression of FGF2 in healthy breast tissue from reduction mammoplasties. In normal tissue, FGF2-tissue is detected in the basal tissue layer, in blood vessels, and in fibroblasts as may be seen below (PMID: 9223382).

[REDACTED]

Regarding the referee's comment on the intensity of the FGF2 in regions of lower compression; Measuring the local variation in physical stiffness in formalin-fixed and paraffin embedded tissue sections is not feasible with current methodologies. One could try to identify low-stiffness related areas from spatial sequencing data, however, there are currently no reliably validated gene sets for low matrix stiffness, and establishing these signatures would be a different project, outside the scope of this study. Moreover, as current spatial sequencing data contains often several cell types per spot, estimating reliably the tumor cell related stiffness would be highly inaccurate and thus we have not been considering this type of analyses.

Minor comments:

Figure 1c-d. Please comment on what the color gradient is intended to quantify.

- In the Figures 1c-d the color gradient quantifies the relative cell number, alongside the dot size. This has been added to the publication.

Figure 1g. The scaling is different for each image making it hard to compare.

- We thank the reviewer from this comment. However, since 3D cultures consists of fragments of varying sizes, we prioritized high-quality images, which is why the scaling differs for each image.

Figure 2. Legend typo ... "patients were cultured in soft matrices" should be PDECS were cultured in ...

- We thank the referee for pointing this out. We have corrected the mistake.

Fig 4C is labeled in the figure as CD86 but referenced in the text to be a measurement of Heparin

- We thank the referee for pointing this out. We have corrected the mistake.

Fig 3G . The text states the figure compares mRNA expression of several genes in PDECs that were uncultured, cultured in stiff matrices, or in soft matrices. The figure does not display results for uncultured cells.

- We thank the referee for pointing this out. The image contains also the uncultured sample, and hence we have corrected the labeling.

REVIEWER COMMENTS

Reviewer #1 (Remarks to the Author):

The authors respond to all questions raised.

There are still 2 pending questions that were not properly answered here below.

Pending questions:

1. Authors' original response to reviewer 1 (1st revision):

The single cell sequencing was performed on two patient samples, P981T and P982T. The results showed depletion of M1 signatures in both patient samples. The P981T demonstrated also upregulation of M2 signatures whereas the P982T sample showed decrease in relative M1 to M2 signature, but we could not find significant M2 signature in P982T in the scRNASeq data. This could be due to resolution issues related to lower immune cell count in P982T sample (see Figure 2B). To confirm the observed upregulation of M2 macrophages in soft matrix cultures, we analyzed macrophage phenotypes from three additional patient samples with flow cytometry. The patient samples were grown as PDECs in soft and stiff (PG) for three days before analyzing the macrophage markers. The new Figure 2J shows the upregulation of CD163 and CD206 marker of M2 macrophage in all the patients. The new data have been added to the revised manuscript as Figure 2 J and new text has been added accordingly (rows 226-233).

Reviewer 1 response:

The authors have now 3 new patients included. The data from the 3 patients do not show statistically significant differences in terms of expression on CD206 and CD163. Why there is no analysis of CD8+ T cells in the all three patients as provided for macrophages in order to appreciate if there is any statistically differences?

1. Authors' response (2nd revision):

The results in 2G and 2J are from the same experiment. The difference in CD8+ T cells between the soft and stiff matrix was statistically significant, as indicated by the line between the circles. For clarity, we have included a note on the significance in the figure legend. The results in the J figure were not significant due to the big variation between the individual patient responses. To address the reviewer's question, we increased the number of analyzed patient samples from the

original 3 to 8 in Figure 2 J. The new data confirm our earlier notion, showing higher levels of CD206 and CD163 in the soft matrix in comparison to stiff matrix. Only one sample did not follow the same trend (P1825T). The observed differences were statistically significant.

For the revised version, new Figure 2 J has been added, and we have modified text and figure legend. Text: “In the scRNASeq analysis, we used uncultured sample as a control for the soft NC cultures. To further explore macrophage phenotypes in stiff versus soft PDEC cultures, we cultured tumors from 8 individual patients in stiff and soft PG matrices and analyzed the median expression of two M2 markers, CD206 and CD163, in the macrophage population of the PDECs (Fig. 2J). In 7 samples, both CD163 and CD206 expressing cell populations were higher in the soft PG matrix compared to the stiff PG matrix ($p= 0.04$ for CD206 and 0.043 for CD163 (Fig. 2J). One sample (P1825T) did not follow this trend for reasons that are unclear.”

2. Authors’ original response to reviewer 1 (1st revision):

To quantify the phenotype of PBMC-derived macrophages after FGF2 + heparin treatment, we assembled a qRT-PCR primer panel, containing a set of M1 and M2 markers, including Arg1, iNos2, and CD80 pointed out by the reviewer. The panel was based on publications (PMID: 29961579, PMID: 35325594, PMID: 37322116). The mRNA expression of CD80 was not significantly altered between the control and FGF2 + heparin treated macrophages. As we did not detect Arg1 or iNos with any of the primers used, we also performed the western blot

analysis for these markers and concluded that PBMC-derived macrophages express low or undetectable levels of Arg1 and iNOS. The negative western blot results are shown below and intended for the reviewer only. Thus, we are unable to conclusively define whether these markers are significantly altered between the control and FGF2 + heparin treated macrophages. However, from our qRT-PCR panel we were able to conclude that M2-related macrophage marker (CCL17) was upregulated with FGF2 + heparin treatment, while M1-related macrophage marker (CXCL9) was downregulated. We hope that the CCL17 and CXCL9 macrophage polarization markers, which were consistently changed in three PBMC samples, provide sufficient evidence to support M2-polarization of FGF2 + Heparin treated macrophages. The reason why we did not observe changes or expression of some of the markers pointed out by the reviewer is most likely due to differences in the macrophage origin. In vitro differentiated macrophages from PBMCs are slightly different from tissue derived macrophages as they originate from blood monocytes and not from the primitive erythro-myeloid progenitors (EMPs) as the tissue derived macrophages (PMID: 24854589, PMID: 31189095). Therefore, some of the most common M2-markers and TAM-markers may be differentially expressed between the PBMC-derived and tissue-derived macrophages.

Reviewer 1 response:

The authors claim that In vitro differentiated macrophages from PBMCs do not express M2 markers. They do not express it if there are not polarized into an M2 phenotype. Therefore a control (using anti-IL4 and IL-13) should be used to compare the results with FGF2+Heparin treated macrophages.

Authors' response:

We thank the reviewer for the comment. IL4 + IL13 cytokines polarize M0 macrophages to M2a subtype, with CD163^{low}/CD206^{high} phenotype (PMID: 35286356, PMID: 35522300, PMID: 39464890). As expected, IL4 + IL13 lowered the CD163 expression and slightly increased the CD206 in treated M0 macrophages, indicating M2a polarization. However, FGF2 + heparin treatment had different effects on CD163 and CD206 expression. The FGF2 + heparin increased the CD163 and lowered the CD206 expression. While the phenotype of FGF2 + heparin differentiated macrophages requires more thorough characterization in future studies, the appearance of the CD163^{high} phenotype together with secretion of immunosuppressive cytokines IL10 and TGFβ resemble M2c macrophages characterized by a CD163^{high}/CD206^{low} phenotype (PMID: 35286356, PMID: 37530555, PMID: 36845095). M2c macrophages mediate

phagocytosis, immunosuppression, angiogenesis, and development of tissue fibrosis, and they are polarized with IL10, TGF β , and glucocorticoids (PMID: 35286356, PMID: 37530555, PMID: 36845095).

We also observed an increase in CD163 positivity in the tissue macrophages residing in soft PDEC cultures. CD206 expression was higher in these cultures. However, given that CD206 is predominantly expressed on tissue macrophages and on immature dendritic cells present in breast tumors (PMID: 7629501; PMID: 10562317), its expression pattern can differ between *in vitro* monocyte-derived macrophages and *ex vivo* tissue macrophages due to differences in the local microenvironment and differentiation cues. These distinctions should be considered when interpreting CD206 trends in different contexts.

Our new results are in line with existing literature where M0 and different subclasses of M2 macrophages express different amounts of CD163 and CD206 based on the cytokines used for polarization. In our setup, M-CSF increases scavenger receptor CD163 expression on M0, whereas the additional IL4 supplementation in M2a decreases the overall CD163 for those cells (PMID: 35522300, PMID: 22900885, PMID: 16297160). Altogether, compared to *in vitro* M2a macrophages as a reference point, we propose that the FGF2 + heparin macrophage from monocyte-derived macrophages resemble M2c macrophages characterized by a CD163^{high}/CD206^{low} phenotype (PMID: 35286356).

To accommodate the new results, we have made the following changes to the manuscript.

New Figure 4 B and Supplementary Figure 6 D.

Fig 4B

SFig 6D

Revised results:

“Macrophages were treated with IFN γ and lipopolysaccharide (LPS) to polarize them into M1 and for M2 polarization, the M0 macrophages were treated with IL4 + IL13 (PMID:25035950). As expected, IFN γ + LPS decreased the CD163 and CD206 expression, consistent with polarization to M1. Furthermore, IL4 + IL13 treatment decreased CD163, while increasing expression of CD206-positive cells. The resulting CD163^{low}; CD206^{high} phenotype is consistent with M2a subtype of M2 macrophages (PMID: 35286356, PMID: 35522300, PMID: 39464890). Intriguingly, FGF2 + heparin increased the CD163 and lowered the CD206 expression, resembling M2c macrophages characterized by a CD163^{high}/CD206^{low} phenotype (Fig 4B, SFig 6D). M2c macrophages mediate phagocytosis, immunosuppression, angiogenesis, and the development of tissue fibrosis, and they are polarized with IL10, TGF β , and glucocorticoids (PMID: 35286356, PMID: 37530555, PMID: 36845095).” (rows 309-319)

Revised discussion:

We show that FGF2 + heparin treatment increased CD163 and lowered the CD206 expression in M0 macrophages. While the phenotype of FGF2 + heparin differentiated macrophages requires more thorough characterization in further studies, the appearance of the CD163^{high} phenotype together with secretion of immunosuppressive cytokines IL10 and TGF β resemble M2c macrophages characterized by a CD163^{high}/CD206^{low} phenotype (PMID: 35286356, PMID: 37530555, PMID: 37081874).” (rows 431-435)

Edited Figure Legend 4 B:

Median fluorescent intensity of CD163 in the M0 macrophages cultured with FGF2, heparin + FGF2, heparin overnight and compared with LPS + IFN γ (M1) and IL-4 + IL-13 (M2a) treated macrophages (N=4-12). (rows 882-884)

Reviewer #2 (Remarks to the Author):

The authors have done their best to address my queries/comments/concerns. I am very satisfied with their effort and improvement of the manuscript.